# Time series forecasting from partial observations via Non-negative Matrix Factorization

## Abstract

In modern time series problems, one aims at forecasting multiple times series with possible missing and noisy values. In this paper, we introduce the Sliding Mask Method (SMM) for forecasting multiple nonnegative time series by means of nonnegative matrix completion: observed noisy values and forecast/missing values are collected into matrix form, and learning is achieved by representing its rows as a convex combination of a small number of nonnegative vectors, referred to as the archetypes. We introduce two estimates, the mask Archetypal Matrix factorization (mAMF) and the mask normalized Nonnegative Matrix Factorization (mNMF) which can be combined with the SMM method. We prove that these estimates recover the true archetypes with an error proportional to the noise. We use a proximal alternating linearized method (PALM) to compute the archetypes and the convex combination weights. We compared our estimators with state-of-the-art methods (Transformers, LSTM, SARIMAX...) in multiple time series forecasting on real data and obtain that our method outperforms them in most of the experiments.

## 1 Introduction

This article investigates forecasting multiple nonnegative times series with missing or noisy entries. We observe $N \geq 1$ time series $\mathbf{M}^{(1)}, \ldots, \mathbf{M}^{(N)} \in \mathbb{R}^T$ over a period of time of length $T \geq 1$ and we would like to forecast the next $F \geq 1$ future values by means of matrix completion, see Figure 1. We define a matrix $\mathbf{M} \in \mathbf{R}^{N \times T}$ whose rows are denoted by $\mathbf{M}^{(i)}$ and columns by $\mathbf{M}_j$. The forecast columns are $\hat{\mathbf{M}}_{T+k}$ for $k = 1, \ldots, F$.

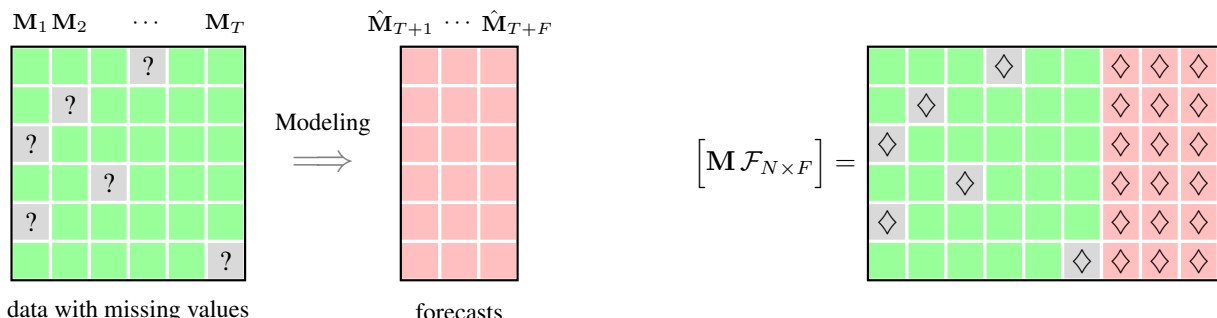

Figure 1: [Left] Consider noisy multiple time series observations (green) with possible missing entries (question mark) from $N \geq 1$ time series and their $F$ forecast values in red. [Right] Matrix completion problem under consideration: missing and forecast values $\mathcal{F}_{N \times F}$ (gray and red diamonds) are not observed.

The matrix completion problem depicted in Figure 1 is ill-posed; standard low-rank techniques cannot recover the missing future values without structural assumptions. To address this, we introduce a deterministic transformation $\boldsymbol{\Phi}$ based on a sliding window approach, referred to as the *Sliding Mask Method (SMM)*.

- **Stride Parameter ($P$):** We define a scalar $P \geq 1$, which determines the stride (or step size) of the sliding window. While often chosen to match a suspected seasonality in the data (e.g., $P = 7$ for weekly cycles), $P$ is a user-defined hyperparameter and does not strictly require intrinsic signal periodicity.

- **Block Construction:** We partition the total time horizon $T + F$ into $B$ blocks of length $P$. To ensuring integer division, we pad the end of the time series with at most $P - 1$ placeholder columns (which are treated as unobserved). Thus, $B = \lceil (T + F)/P \rceil$.

- **Sliding Window Transformation:** The output matrix is constructed by stacking windows of length $WP$, where $W$ is the number of consecutive sub-blocks per window. This transforms the original $N \times (T + F)$ matrix into a larger matrix where rows represent local time-segments.

- **Forecasting as Completion:** We assume $WP > F$. The resulting matrix structure (see Figure 2) places the values to be forecasted into a specific sub-block structure. This converts the temporal forecasting problem into a structured matrix completion problem.

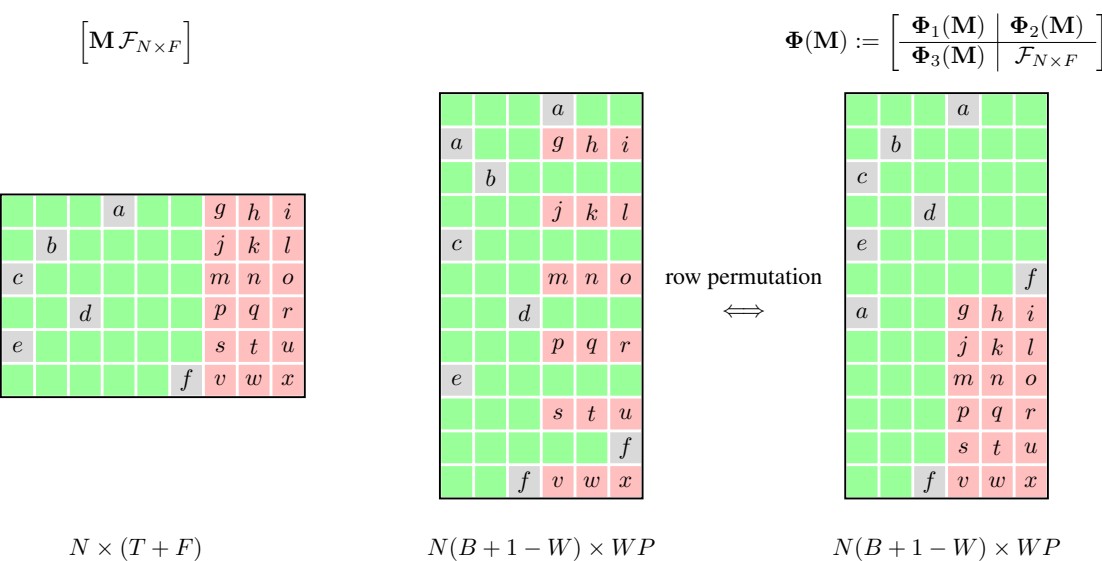

Figure 2: Given a matrix of size $N \times (T + F)$ of observations $\mathbf{M}$ and forecasts $\mathcal{F}_{N \times F}$, the output sliding mask matrix is composed of 4 blocks denoted by $\mathbf{\Phi}_1(\mathbf{M})$, $\mathbf{\Phi}_2(\mathbf{M})$, $\mathbf{\Phi}_3(\mathbf{M})$, and $\mathcal{F}_{N \times F}$, this latter being of size $N \times F$ with the $F$ last columns of the input matrix. In this example, we consider a periodicity of $P = 3$, giving $B = 3$ sub-blocks per row of the input matrix and we gather $W = 2$ consecutive sub-blocks in an output row. To ease readability, we denote by $a, \ldots, f$ the missing values and by $g, \ldots, x$ the values to forecast. After row permutation, we obtain the output SMM. The matrix $\mathbf{\Phi}_1(\mathbf{M})$ (resp. $\mathbf{\Phi}_2(\mathbf{M})$, $\mathbf{\Phi}_3(\mathbf{M})$) is a sub-matrix of size $N(B - W) \times (WP - F)$ (resp. $N(B - W) \times F$, $N \times (WP - F)$) of the input matrix $\mathbf{M}$.

We define our matrix completion problems on the observation matrix $\mathbf{X}$ using the mask operator $\mathcal{T}$ defined below.

**Definition 1 (Observation Matrix X)** *Let $n := N(B+1-W)$ and $p := WP$. Given the input time series matrix $\mathbf{M}$, we apply the transformation $\mathbf{\Phi}$ to obtain the full (ground truth) matrix. The* observation matrix $\mathbf{X} \in \mathbb{R}^{n \times p}$ *is defined by retaining the known past values and setting unknown future (forecast) entries and missing data to zero.*

**Definition 2 (Mask Operator $\mathcal{T}$)** *Let $\Omega$ be the set of indices $(i, j)$ corresponding to observed values in the input data. We define the linear mask operator $\mathcal{T} : \mathbb{R}^{n \times p} \to \mathbb{R}^{n \times p}$ as:*

$$[\mathcal{T}(\mathbf{N})]_{i,j} = \begin{cases} \mathbf{N}_{i,j} & \text{if } (i, j) \in \Omega \\ 0 & \text{otherwise} \end{cases} \tag{1}$$

*Consequently, our data consistency constraint is written as $\mathcal{T}(\mathbf{N}) = \mathbf{X}$. Note that unobserved entries in $\mathbf{X}$ are zero, and $\mathcal{T}$ forces the approximation $\mathbf{N}$ to match $\mathbf{X}$ only on the observed support $\Omega$.*

We introduce two factorization formulations. The first is the standard Normalized NMF, while the second (Archetypal) imposes stronger convexity constraints, often leading to more interpretable "archetypes" robust to outliers.

**Definition 3 (Mask Normalized NMF (mNMF))** *We solve for a completion* $\mathbf{N} \approx \mathbf{WH}$ *minimizing the reconstruction error only on observed entries:*

$$\min_{\mathbf{W},\mathbf{H}} \left\{ \|\mathcal{T}(\mathbf{X} - \mathbf{WH})\|_F^2 \right\} \quad s.t. \quad \mathbf{W} \geq \mathbf{0}, \mathbf{H} \geq \mathbf{0}, \mathbf{W1} = \mathbf{1}. \tag{mNMF}$$

**Definition 4 (Mask Archetypal Matrix Factorization (mAMF))** *We relax the exact factorization but enforce that the factors* $\mathbf{H}$ *(archetypes) lie within the convex hull of the data (approximated by* $\mathbf{VX}$*). This is formulated as:*

$$\min_{\mathbf{W},\mathbf{H},\mathbf{V}} \left\{ \|\mathcal{T}(\mathbf{X} - \mathbf{WH})\|_F^2 + \lambda\|\mathbf{H} - \mathbf{VX}\|_F^2 \right\} \tag{mAMF}$$

*subject to* $\mathbf{W}, \mathbf{V} \geq \mathbf{0}$*, and column-stochastic constraints* $\mathbf{W1} = \mathbf{1}, \mathbf{V1} = \mathbf{1}$*.*

**Definition 5 (Normalization, nonnegative and archetype basis)** *The matrix* $\mathbf{W}$ *satisfies the constraint* $\mathbf{W} \geq \mathbf{0}$ *and* $\mathbf{W1} = \mathbf{1}$*, this being later referred to as normalization. Its rows* $(w_{i,1}, \ldots, w_{i,K})$ *are convex combination weights and each row of* $\mathbf{WH}$ *is a convex combination of the* $K$ *rows of* $\mathbf{H}$*. The matrix* $\mathbf{H}$*, referred to as the* nonnegative basis *(resp. archetype basis) in* (mNMF) *(resp.* (mAMF)*).*

*In* (mAMF) *as* $\lambda \to \infty$*, the* archetypes *(defined as the* $K$ *rows of* $\mathbf{H}$*) are forced to lie in the convex combination of* $\mathbf{N}$ *by means of the matrix* $\mathbf{V}$*. Since* $\mathbf{N}$ *is a completion of the observation matrix* $\mathbf{X}$ *by the mask operator* $\mathbf{N} = \mathcal{T}(\mathbf{X})$*, the matrix* $\mathbf{V}$ *can be interpreted as the convex combination weights of the decomposition of the archetypes onto the rows of the observation matrix* $\mathbf{X}$ *and hence we recover the method of* Cutler & Breiman *(1994).*

Hence we get the following decomposition of the $i^{th}$ row of $\mathbf{WH}$,

$$(\mathbf{WH})^{(i)} = \sum_{k=1}^{K} w_{i,k}\mathbf{H}_k \,. \tag{2}$$

Once solved, the above matrix problems give forecast values to the original forecasting problem of time series by means of matrix $\hat{\mathbf{M}}$ defined below.

**Definition 6 (Forecasts of the original problem)** *Forecasts* $\hat{\mathbf{M}} \in \mathbb{R}^{N \times F}$ *are given by the bottom right* $N \times F$ *submatrix of* $\mathbf{WH}$*, namely* $\hat{\mathbf{M}}$ *is the bottom right red block in Figure* 2 *and it is the same block as* $\mathcal{F}_{N \times F}$ *in the original forecasting problem (letters* $g$ *to* $x$ *in Figure* 2*), hence* $\hat{\mathbf{M}}$ *can be interpreted as forecast values.*

## 1.1 Mask nonnegative matrix completion statistical guarantees

**Our goal is to solve the following nonnegative matrix completion problem:** We observe a matrix $\mathbf{X} \in \mathbb{R}^{n \times p}$ containing the multiple time values, given by the transformation presented in Figure 2. The missing values and the forecast values are arbitrarily set to zero, see (1). This choice is *not restrictive* since the values of $\mathbf{X}$ corresponding to the missing and forecast entries are *not observed* and our study is *insensitive* to the values of these entries. Our target is defined by the following best approximation of $\mathbf{X}$ through the mask operator $\mathcal{T}$.

**Definition 7 (Best normalized nonnegative rank $K$ approximation of $\mathbf{X}$)** *Given a nonnegative rank* $K$*, we define the following best approximation of* $\mathbf{X}$*:*

$$\mathbf{X}_0 := \mathbf{W}_0\mathbf{H}_0 \in \arg \min_{\substack{\mathbf{W}_0\mathbf{1}=\mathbf{1} \\ \mathbf{W}_0 \geq \mathbf{0}, \, \mathbf{H}_0 \geq \mathbf{0}}} \left\{ \left\|\mathbf{X} - \mathcal{T}(\mathbf{W}_0\mathbf{H}_0)\right\|_F^2 \right\}, \tag{3}$$

*where* $\mathbf{W}_0 \in \mathbb{R}^{n \times K}$ *and* $\mathbf{H}_0 \in \mathbb{R}^{K \times p}$*. As* $K$ *grows, the approximation error* $\|\mathbf{X} - \mathcal{T}(\mathbf{W}_0\mathbf{H}_0)\|_F$ *decreases. The matrix* $\mathbf{X}_0$ *is referred to as the best normalized nonnegative rank* $K$ *approximation of* $\mathbf{X}$*.*

The goal is to recover the matrices $\mathbf{W}_0$ (weights) and $\mathbf{H}_0$ (archetypes) from the observation matrix $\mathbf{X}$. The observation can be written as

$$\mathbf{X} = \mathcal{T}(\mathbf{X}_0) + \mathbf{F}\,, \tag{4}$$

where $\mathbf{F}$ is some additive error term, referred to as the noise.

**Contributions**   Sliding Mask Method (SMM) outputs the forecast values and it can be viewed as a nonnegative matrix completion algorithm under low nonnegative rank assumption. This framework raises two issues. A first question is the uniqueness of the decomposition, also referred to as *identifiability* of the model. In Theorem 14, we introduce a new condition that ensures uniqueness from partial observation of the target matrix. An other challenge, as pointed out by Vavasis (2009) for instance, is that solving *exactly* NMF decomposition problem is NP-hard. Nevertheless NMF-type problems can be solved efficiently using (accelerated) proximal gradient descent method Parikh & Boyd (2013) for block-matrix coordinate descent in an *alternating projection scheme*, *e.g.*, Javadi & Montanari (2020a) and references therein. We rely on these techniques to introduce algorithms inputting the forecast values based on NMF decomposition, see Section 3. Theorem 17 complements the theoretical analysis by proving the robustness of NMF-type algorithms when entries are missing or corrupted by noise. Our main theoretical contributions are as follows:

- A uniqueness decomposition result (Theorem 14) showing that the decomposition $\mathbf{W}_0\mathbf{H}_0$ is unique *given partial observations*, namely

$$\text{If} \quad \mathcal{T}(\mathbf{WH}) = \mathcal{T}(\mathbf{W}_0\mathbf{H}_0) \quad \text{then} \quad (\mathbf{W}, \mathbf{H}) \equiv (\mathbf{W}_0, \mathbf{H}_0)\,, \tag{$\mathbb{P}_\mathrm{u}$}$$

  where $\equiv$ means up to positive scaling and permutation[1].

- A robustness result (Theorem 17 and Corollary 18) showing that (mNMF) and (mAMF) recover $\mathbf{H}_0$ and $\mathbf{W}_0$ with an error proportional to $\|\mathbf{F}\|_F$ (hence we recover $\mathbf{X}_0$ with the same precision).

Our analysis is completed by an algorithmic and numerical study that

- introduces a Proximal Alternating Linearized Minimization (PALM) method to solve (mAMF) and shows that PALM reaches a stationary point (Theorem 20).

- reports a performance improvement of (mNMF) and (mAMF) against state-of-the-art algorithms on real datasets for RRMSE and RMPE (Table 1). The relative root-mean-squared error (RRMSE) and the relative mean-percentage error (RMPE) are defined by

$$\text{RRMSE} = \frac{\|\mathbf{M}_F - \mathbf{M}_F^\star\|_F}{\|\mathbf{M}_F^\star\|_F}\,, \ \text{RMPE} = \frac{\|\mathbf{M}_F - \mathbf{M}_F^\star\|_1}{\|\mathbf{M}_F^\star\|_1}\,.$$

  where $\mathbf{M}_F^\star$ are the true values and $\mathbf{M}_F$ the forecasts (see Section 4).

**Comments on low rank modeling and periodicity in time series**   Sparse or Low-Rank representations are ubiquitous in applications and well studied in the literature. In our analysis a time series is cut into several smaller $W$ sub-blocks time series with the same length $p = WP$. For instance, observing sales over a period of one year, one can consider 52 weekly time series (one per week). These observations are the rows of our observed matrix $\mathbf{X}$. The normalized nonnegative low rank hypothesis assumes that the $p$-length multiple time series of the dataset can be decomposed as a sum of $K$ basis time series $\mathbf{H}$ plus an error term. Of course, this error term can incorporate the model approximation error as depicted in (3). The $K$ basis time series $\mathbf{H}$ are learned on the entire dataset $\mathbf{X}$. This technique can be seen as dimension reduction, each observation can be summarized as $K$ weights $\mathbf{W}$ such that the resulting convex combination of basis time series (2) is a good approximation of the observation $\mathbf{X}$.

The low rank hypothesis can be interpreted as a periodicity assumption. Indeed, if the time series are exactly periodic with period $p$, then the rank of the data matrix $\mathbf{X}$ is at most $p$. While $P$ is a free parameter, the model's performance

---

[1]if an entry-wise nonnegative pair $(\mathbf{W}, \mathbf{H})$ is given then $(\mathbf{WPD}, \mathbf{D}^{-1}\mathbf{P}^\top\mathbf{H})$ is also a nonnegative decomposition $\mathbf{WH} = \mathbf{WPD} \times \mathbf{D}^{-1}\mathbf{P}^\top\mathbf{H}$, where $\mathbf{D}$ scales and $\mathbf{P}$ permutes the columns (resp. rows) of $\mathbf{W}$ (resp. $\mathbf{H}$)

| Algorithms | mAMF | | mNMF | | RFR | | EXP | | SARIMAX | | LSTM | | GRU | |
|---|---|---|---|---|---|---|---|---|---|---|---|---|---|---|
| Metrics | RRMSE | RMPE | RRMSE | RMPE | RRMSE | RMPE | RRMSE | RMPE | RRMSE | RMPE | RRMSE | RMPE | RRMSE | RMPE |
| daily electricity | 14.42% | 36.85% | 15.86% | 46.66% | 12.16% | 47.78% | 11.25% | 43.83% | 9.85% | 43.16% | 12.42% | 46.49% | 12.03% | 45.90% |
| weekly electricity | 14.80% | 17.50% | 11.09% | 13.79% | 7.25% | 8.61% | 10.07% | 7.98% | 9.05% | 7.42% | 27.85% | 15.64% | 26.04% | 15.92% |
| gas | **21.71%** | **18.55%** | 37.46% | 42.79% | 66.80% | 71.61% | 63.35% | 68.16% | 45.58% | 52.83% | 62.97% | 68.38% | 62.87% | 67.90% |
| Istanbul | 15.67% | 17.80% | **14.18%** | 16.77% | 15.37% | 18.32% | 15.46% | 18.64% | 14.75% | 17.01% | 16.22% | 20.96% | 20.01% | 26.87% |
| ETTh1 | **10.24%** | 15.23% | 12.30% | 14.16% | 12.96% | 17.98% | 12.37% | 13.65% | 13.36% | 15.94% | 14.86% | 18.78% | 14.71% | 18.85% |
| ETTh2 | 9.42% | 13.07% | **4.87%** | **6.66%** | 6.47% | 7.60% | 14.06% | 13.67% | 12.76% | 13.03% | 14.17% | 13.75% | 14.44% | 14.36% |
| ETTm1 | 10.12% | 15.22% | 9.94% | **12.25%** | 12.81% | 17.42% | 11.45% | 14.20% | 12.29% | 16.45% | 13.39% | 17.96% | 14.13% | 18.63% |
| ETTm2 | 8.19% | 11.65% | **5.08%** | 7.41% | 5.81% | **7.16%** | 13.18% | 12.88% | 13.16% | 12.95% | 14.29% | 13.89% | 14.46% | 14.03% |
| electricity1 | **6.59%** | 13.17% | 11.11% | 15.28% | 12.75% | 16.09% | 38.27% | 34.44% | >100.00% | >100.00% | 8.51% | 10.13% | 7.19% | **9.61%** |
| electricity2 | **8.09%** | 16.82% | 8.82% | **12.17%** | 12.05% | 15.67% | 47.40% | 40.36% | 43.38% | 38.98% | 9.05% | 12.83% | 10.30% | 13.28% |
| electricity3 | 10.57% | 13.95% | 12.43% | 14.04% | 12.45% | 14.14% | 40.37% | 33.48% | 37.05% | 33.01% | 10.70% | 11.62% | 10.70% | 11.02% |
| electricity4 | 11.02% | 24.30% | 25.07% | 29.71% | 23.16% | 19.50% | 54.42% | 43.63% | 63.08% | 46.59% | 12.18% | 13.88% | **9.53%** | **11.05%** |
| electricity5 | 9.52% | 19.05% | **7.72%** | **15.48%** | 25.96% | 26.84% | 56.76% | 49.31% | * | * | 21.92% | 28.79% | 20.73% | 27.02% |
| electricity6 | 10.11% | 17.04% | 14.30% | 18.62% | 13.81% | 16.26% | 51.87% | 37.35% | 52.10% | 40.56% | 7.58% | 11.74% | **7.13%** | **10.32%** |
| electricity7 | **8.34%** | 16.75% | 37.49% | 30.03% | 29.51% | 22.96% | 53.00% | 45.95% | * | * | 17.74% | 14.72% | 16.55% | **14.19%** |
| electricity8 | **10.03%** | 17.49% | 23.81% | 20.59% | 19.33% | 17.98% | 36.83% | 40.65% | 38.54% | 41.23% | 12.16% | **15.73%** | 13.89% | 17.08% |
| electricity9 | 19.45% | 38.90% | 21.15% | 41.72% | 18.18% | 37.53% | 35.90% | 38.65% | >100.00% | >100.00% | 18.00% | 37.77% | 18.80% | 38.21% |
| electricity10 | **5.13%** | 12.53% | 5.40% | 11.29% | 12.11% | 13.42% | 33.88% | 34.89% | 36.55% | 38.92% | 7.66% | 10.25% | 7.77% | 9.94% |
| synthetic1 | 6.40% | 9.30% | 5.81% | **8.01%** | 5.79% | 9.44% | **5.73%** | 9.32% | 5.76% | 8.81% | 6.67% | 11.87% | 6.77% | 11.89% |
| synthetic2 | 19.04% | 20.28% | 20.35% | 25.30% | 17.12% | 20.24% | 21.09% | 25.87% | 28.41% | 35.53% | 21.55% | 29.04% | 21.48% | 28.94% |
| low-noise | **0.10%** | **0.26%** | 0.10% | 0.26% | 8.65% | 22.76% | 16.97% | 48.02% | 0.19% | 0.33% | 16.52% | 46.03% | 16.76% | 46.46% |
| medium-noise | 2.41% | 5.23% | **1.92%** | **4.81%** | 8.42% | 21.95% | 15.94% | 44.25% | 1.97% | 4.94% | 15.66% | 42.68% | 15.67% | 42.98% |
| high-noise | 12.69% | 28.04% | **10.39%** | **26.43%** | 11.73% | 30.26% | 13.02% | 33.27% | 15.24% | 27.37% | 13.03% | 33.67% | 12.97% | 33.47% |

| Algorithms | BasisFormer | | Autoformer | | iTransformer | | PatchMLP | | TimeMixer | |
|---|---|---|---|---|---|---|---|---|---|---|
| Metrics | RRMSE | RMPE | RRMSE | RMPE | RRMSE | RMPE | RRMSE | RMPE | RRMSE | RMPE |
| daily electricity | **7.56%** | 6.64% | 39.65% | 80.25% | 28.85% | 61.63% | 8.30% | **6.39%** | 32.15% | 68.83% |
| weekly electricity | 8.76% | 9.07% | 41.34% | 80.93% | 37.27% | 75.07% | **6.99%** | 7.57% | 44.07% | 82.74% |
| gas | 57.45% | 52.10% | 99.37% | >100.00% | >100.00% | >100.00% | – | – | – | – |
| Istanbul | 14.83% | **12.54%** | 80.00% | 89.26% | >100.00% | >100.00% | 18.77% | 18.93% | >100.00% | >100.00% |
| ETTh1 | 14.57% | **13.61%** | 52.37% | 34.35% | 42.74% | 30.95% | 19.94% | 25.41% | 36.92% | 24.39% |
| ETTh2 | 54.66% | 53.66% | 66.85% | 73.31% | 65.10% | 69.41% | 53.59% | 53.68% | 56.72% | 60.34% |
| ETTm1 | 13.58% | 12.31% | 49.72% | 34.94% | 36.77% | 28.11% | 21.28% | 26.48% | 36.94% | 23.70 |
| ETTm2 | 55.52% | 54.95% | 59.79% | 66.20% | 64.78% | 68.69% | 53.89% | 53.38% | 57.21% | 61.53% |
| electricity1 | 26.93% | 28.19% | 49.69% | 83.08% | 36.84% | 80.29% | 11.97% | 12.29% | 58.63% | 91.57% |
| electricity2 | 35.38% | 39.73% | 49.00% | 52.77% | 38.62% | 47.73 | 10.42% | 11.48% | 56.70% | 55.77% |
| electricity3 | 34.30% | 37.25% | 52.13% | 90.29% | 38.04% | 68.80% | **9.13%** | **9.12%** | 62.73% | 79.92% |
| electricity4 | 39.42% | 40.66% | 46.88% | 81.34% | 44.36% | 85.28% | 15.52% | 15.80% | 54.53% | 90.54% |
| electricity5 | 46.22% | 49.60% | 49.16% | 43.74% | 55.43% | 44.32% | 13.94% | 13.36% | 53.67% | 42.78% |
| electricity6 | 45.50% | 46.86% | 50.51% | 73.42% | 46.08% | 79.51% | 18.58% | 17.00% | 50.77% | 75.83% |
| electricity7 | 40.17% | 43.20% | 86.87% | >100.00% | 84.60% | >100.00% | 16.72% | 17.27% | 85.16% | >100.00% |
| electricity8 | 30.64% | 30.99% | 62.89% | 98.00% | 47.05% | 81.41% | 13.62% | 14.21% | 62.71% | 95.12% |
| electricity9 | 34.88% | 35.85% | 40.79% | 27.79% | 25.32% | 26.44% | **11.32%** | **11.08%** | 54.67% | 29.59% |
| electricity10 | 29.78% | 31.79% | 54.90% | >100.00% | 35.75% | >100.00% | 8.15% | **8.51%** | 66.22% | >100.00% |
| synthetic1 | 8.06% | 12.32% | > 100.00% | >100.00% | >100.00% | >100.00% | 6.52% | 11.74% | 96.27% | 94.14% |
| synthetic2 | 31.32% | 47.88% | 59.75% | 83.66% | 60.08% | >100.00% | **1.33%** | **7.53%** | 55.61% | 69.21% |
| low-noise | 23.71% | 51.94% | 95.23% | 80.40% | 7.18% | 13.73% | 10.28% | 27.49% | >100.00% | >100.00% |
| medium-noise | 21.68% | 48.48% | 83.34% | 73.79% | 16.49% | 30.96% | 9.66% | 27.36% | >100.00% | >100.00% |
| high-noise | 18.44% | 45.77% | >100.00% | >100.00% | 94.25% | >100.00% | 13.49% | 39.92% | >100.00% | >100.00% |

Table 1: Comparison of RRMSE and RMPE metrics. Best results in **bold**, second best underlined. Note: mAMF outperforms standard baselines on datasets with clear local recurring structures (e.g., daily electricity).

depends on $P$ aligning with a quasi-periodic, low-rank structure in the data. Our experiments in Section 4 show that this approach is effective on real-world datasets. In practice, time series are not exactly periodic, but they can be approximated as a sum of few periodic components plus some noise. This is the rationale behind Fourier analysis and wavelet analysis for time series. The low rank hypothesis can be seen as a nonnegative and adaptive generalization of Fourier analysis where the basis time series $\mathbf{H}$ are learned from data.

The relevance of such a hypothesis on real data cannot be proven beforehand. Our numerical study on real data shows that we improve results in prediction, better than standard methods in time series analysis: Seasonal AutoRegressive Integrated Moving Average with eXogenous variables model (SARIMAX), EXPonential moving average (EXP), Random Forest Regressor (RFR), Long Short-Term Memory (LSTM), Gated Recurrent Units (GRU), BasisFormer (Attention-based Time Series Forecasting with Learnable and Interpretable Basis). It suggests that the low rank assumption is reasonable for the datasets studied in the paper.

**Data Reweighting and Overlapping Windows**   The construction of the observation matrix $\mathbf{X}$ involves sliding a window of length $WP$ with a stride of $P$. When $WP > P$, the windows overlap, causing specific time steps to appear in multiple rows of $\mathbf{X}$.

While this introduces a form of data reweighting—where central data points are sampled more frequently than boundary points—this redundancy is intentional. It acts as a deterministic data augmentation strategy that enforces *shift invariance* in the learned archetypes. By presenting the same temporal transition in different columns of the matrix, the algorithm learns robust motifs that are not artifacts of the specific grid alignment. Our empirical results suggest this overlapping strategy stabilizes the factorization, particularly for datasets with weak periodicity, by artificially increasing the number of training samples for the local patterns.

**Selection of Rank** $K$    The nonnegative rank $K$ is a critical hyperparameter governing the model complexity. We select $K$ using a time-based cross-validation strategy. We designate a portion of the historical training data as a validation set (mimicking the forecast block structure). We grid-search $K$ (e.g., $K \in \{4, \dots, 30\}$) and select the value that minimizes the validation Root Mean Squared Error (RMSE) before retraining on the full dataset.

## 1.2    Notation

To ensure clarity, we define our notation early. We denote scalars by lowercase letters (e.g., $x$), vectors by bold lowercase letters (e.g., $\mathbf{x}$), and matrices by bold uppercase letters (e.g., $\mathbf{X}$).

- The input time series matrix is $\mathbf{M} \in \mathbb{R}^{N \times T}$.

- The transformed observation matrix (via the sliding mask) is $\mathbf{X} \in \mathbb{R}^{n \times p}$.

- The ground truth target matrix is denoted by $\mathbf{X}_0$.

- Factor matrices are $\mathbf{W}$ (weights) and $\mathbf{H}$ (archetypes).

- The mask operator is denoted by $\mathcal{T}(\cdot)$, where $\mathcal{T}(\mathbf{N})$ retains entries corresponding to observed values and zeros out missing/forecast entries.

We use $\mathbb{R}_+^{n \times p}$ to denote the set of non-negative $n \times p$ matrices. The Frobenius norm is denoted by $\| \cdot \|_F$. For a comprehensive list of symbols, we refer the reader to Table 2.

| Symbol | Description |
|---|---|
| $N$ | Number of time series |
| $T$ | Length of historical data |
| $F$ | Length of forecast horizon |
| $P$ | Stride parameter (periodicity) |
| $\mathbf{M}$ | Raw time series matrix ($N \times T$) |
| $\mathbf{\Phi}$ | Sliding window transformation operator |
| $\mathbf{X}$ | Observation matrix after transformation ($n \times p$) |
| $\mathbf{X}_0$ | Ground truth low-rank matrix |
| $\mathbf{W}, \mathbf{H}$ | Factor matrices (Weights and Archetypes) |
| $\mathcal{T}$ | Mask operator |
| $K$ | Nonnegative rank |

Table 2: Summary of notations used throughout the paper.

## 1.3    Related Works

Our work intersects with several research areas, including the theory of Nonnegative Matrix Factorization (NMF), its application to time-series analysis, and methods for handling missing data.

**NMF Uniqueness and Our Contribution**    The uniqueness of NMF decompositions is a cornerstone of its theoretical understanding. Foundational work by Thomas (1974) and subsequent analyses by Donoho & Stodden (2004); Laurberg et al. (2008) and Recht et al. (2012) have established conditions under which NMF yields a unique solution, often relying on geometric properties of the data matrix. More recently, conditions such as the Sufficiently

Scattered Condition (SSC) (Huang et al., 2013) have relaxed the requirements for identifiability. Regarding missing data, Ibrahim & Fu (2021) and Gillis (2020) discuss NMF under general block-missing patterns or edge queries. Our work differs by addressing the specific, deterministic "sliding window" missingness pattern induced by the forecasting formulation, rather than random block erasures.

**Time-Series Forecasting Models** The field of time-series forecasting is dominated by statistical and deep learning models. Classical methods like SARIMAX (Seasonal Auto-Regressive Integrated Moving Average with eXogenous variables) assume linear dependencies and specific seasonal patterns. In contrast, deep learning models such as LSTMs (Long Short-Term Memory networks), BasisFormer (Ni et al., 2023), Autoformer (Wu et al., 2021), iTransformer (Liu et al., 2024), PatchMLP (Tang & Zhang, 2025) and TimeMixer (Wang et al., 2024) learn complex, non-linear temporal dependencies from large amounts of data. While these models are state-of-the-art for large-scale series, they often require massive datasets to learn temporal structures and they often operate as "black boxes". Our SMM framework offers a different paradigm: it assumes that time-series segments can be represented as a convex combination of a few learned, interpretable basis vectors (archetypes). This low-rank hypothesis is fundamentally different from the auto-regressive or attention-based mechanisms of other models and provides inherent interpretability, as demonstrated in our experiments (Section 4).

**NMF for Missing Data** The problem of applying NMF to data with missing values is not new and many of these approaches are purely algorithmic. Our primary contribution is that the SMM framework itself—a structured method for converting a time-series forecasting problem into a matrix completion problem. Our theoretical analysis provides guarantees for this specific structure, which general-purpose NMF-for-missing-data algorithms do not offer. The present work assumes block-wise missing structures and provides uniqueness and robustness guarantees in this context, which is novel compared to prior works that often assume random missingness without specific structural patterns. General patterns of missing data are not covered by our analysis and remain an open research question. However, our algorithmic framework can be adapted to other missing data patterns, although without the same theoretical guarantees.

**NMF-based time-series analysis** Our work is distinct from previous NMF-based time-series analysis by Mei et al. (2017); Mei et al. (2018). While Mei et al. also use NMF, their work focuses on recovering high-resolution time series from temporal aggregates (disaggregation) and leveraging side information. For example, they might recover individual household consumption from a neighborhood's total consumption. Our SMM framework is fundamentally different. It operates by creating a matrix of sliding windows from the time series, thereby transforming the forecasting problem into one of finding a low-rank representation of these segments. The goal is to learn archetypal segment patterns for forecasting, not to disaggregate a signal.

Robustness of archetypal analysis have been studied in Javadi & Montanari (2020a) for simplicial polyhedral cone approximation of a dataset, denoted in data matrix form by $\mathbf{X} \in \mathbb{R}^{n \times p}$ in this paper. This paper extends this latter analysis to the case where some data entries might be missing and some data blocks are not observed (forecast, red values in Fig. 2).

## 2 Uniqueness and estimation guarantees

### 2.1 The train and test paradigm, link with forecasting multiple nonnegative time series

The model under consideration is presented in Equations (7). Our goal is to estimate the $K$-best normalized non-negative approximation $\mathbf{X}_0$, defined in Equation (3), from the partial and noisy observation $\mathbf{X}$. We denote by $\mathbf{X}^\star$ is the *mask* of $\mathbf{X}_0$, namely

$$\mathbf{X}^\star := \mathcal{T}(\mathbf{X}_0) = \left[ \begin{array}{c|c} \mathbf{X}_1^\star & \mathbf{X}_2^\star \\ \hline \mathbf{X}_3^\star & \mathbf{0}_{N \times F} \end{array} \right], \tag{5a}$$

where $\mathbf{X}_1^\star \in \mathbb{R}^{(n-N) \times (p-F)}$, $\mathbf{X}_2^\star \in \mathbb{R}^{(n-N) \times F}$, and $\mathbf{X}_3^\star \in \mathbb{R}^{N \times (p-F)}$ are blocks of $\mathbf{X}_0$. Note that $\mathbf{X} = \mathbf{X}^\star + \mathbf{F}$, where $\mathbf{F}$ is the *noise* term, see Equation (4).

These blocks can be be gathered into a *train and test paradigm*. Note that we observe the full sub-matrix $\mathcal{T}_{\text{train}}(\mathbf{X}_0) := [\mathbf{X}_1^\star \ \mathbf{X}_2^\star]$ which we refer to as the training part. We would like to predict (forecast) the $\mathbf{0}_{N \times F}$ block of the sub-matrix

$\mathcal{T}_{\text{test}}(\mathbf{X}_0) := [\mathbf{X}_3^\star \ \mathbf{0}_{N \times F}]$ which we refer to as the test part of our observation $\mathbf{X}$. Looking at Figure 2, we define

$$\mathcal{T}_T(\mathbf{X}_0) := \begin{bmatrix} \mathbf{X}_1^\star \\ \mathbf{X}_3^\star \end{bmatrix} \quad \text{and} \quad \mathcal{T}_F(\mathbf{X}_0) := \begin{bmatrix} \mathbf{X}_2^\star \\ \mathbf{0}_{N \times F} \end{bmatrix}. \tag{5b}$$

Our notation (subscripts $T$ and $F$) stems from the sliding mask method for multiple time series forecast. Note that $\mathcal{T}_T(\mathbf{X})$ gathers all the information observed up to time $T$, and we would like to forecast the $\mathbf{0}_{N \times F}$ block of $\mathcal{T}_F(\mathbf{X})$. Now, we know by design that $\mathbf{X}_0 := \mathbf{W}_0\mathbf{H}_0$. Hence, denoting $\mathbf{H}_0 =: [\mathbf{H}_{0T} \ \mathbf{H}_{0F}]$, and $\mathbf{W}_0^\top =: [\mathbf{W}_{0\text{train}}^\top \ \mathbf{W}_{0\text{test}}^\top]$, we get that

$$\begin{aligned}
\mathcal{T}_{\text{train}}(\mathbf{X}_0) &= \mathbf{W}_{0\text{train}}\mathbf{H}_0, & \mathbf{X}_3^\star &= \mathbf{W}_{0\text{test}}\mathbf{H}_{0T}, \\
\mathcal{T}_T(\mathbf{X}_0) &= \mathbf{W}_0\mathbf{H}_{0T}, & \mathbf{X}_2^\star &= \mathbf{W}_{0\text{train}}\mathbf{H}_{0F}.
\end{aligned} \tag{5c}$$

In light of Figures 1 and 2, the multiple forecasts $\hat{\mathbf{M}}_{T+1}, \dots, \hat{\mathbf{M}}_{T+F}$ can be given a best normalized nonnegative rank $K$ approximation by $\mathbf{W}_{0\text{test}}\mathbf{H}_{0F}$. Observe that an estimation of $\mathbf{W}_{0\text{test}}$ gives the weights learnt on the test sub-matrix while an estimation of $\mathbf{H}_{0F}$ is the forecast of the archetypes, see the decomposition (2).

## 2.2 Uniqueness from partial observations

When we observe the full matrix $\mathbf{X}_0 = \mathbf{W}_0\mathbf{H}_0$, the issue on uniqueness has been addressed under some sufficient conditions on $\mathbf{W}, \mathbf{H}$, *e.g.*, *Strongly boundary closeness* of Laurberg et al. (2008), *Complete factorial sampling* of Donoho & Stodden (2004), and *Separability* of Recht et al. (2012). A necessary and sufficient condition exists as given by the following theorem. We recall that that the $K$-dimensional positive orthant is the set $\{x \in \mathbb{R}^K : x_i \geq 0, \ \forall i \in [K]\}$ and a $K$-simplicial cone is the set which the conic hull of $K$ linearly independent vectors of $\mathbb{R}^K$.

**Theorem 8 (Thomas (1974))** *The decomposition $\mathbf{X}_0 := \mathbf{W}_0\mathbf{H}_0$ is unique up to permutation and positive scaling of columns (resp. rows) of $\mathbf{W}_0$ (resp. $\mathbf{H}_0$)* **if and only if** *the $K$-dimensional positive orthant is the only $K$-simplicial cone verifying $\text{Cone}(\mathbf{W}_0^\top) \subseteq \mathcal{C} \subseteq \text{Cone}(\mathbf{H}_0)$ where $\text{Cone}(\mathbf{A})$ is the cone generated by the rows of $\mathbf{A}$.*

Our first assumption is following.

**Assumption 1** *In the set given by the union of sets:*

$$\{\mathcal{C} \ : \ \text{Cone}(\mathbf{W}_{0\text{train}}^\top) \subseteq \mathcal{C} \subseteq \text{Cone}(\mathbf{H}_0)\} \bigcup \{\mathcal{C} \ : \ \text{Cone}(\mathbf{W}_0^\top) \subseteq \mathcal{C} \subseteq \text{Cone}(\mathbf{H}_{0T})\}, \tag{$\mathbb{A}_1$}$$

*the nonnegative orthant is the only $K$-simplicial cone. Note that this assumption is implied by the following one: In the set*

$$\{\mathcal{C} \ : \ \text{Cone}(\mathbf{W}_{0\text{train}}^\top) \subseteq \mathcal{C} \subseteq \text{Cone}(\mathbf{H}_{0T})\} \tag{$\mathbb{A}_1'$}$$

*the nonnegative orthant is the only $K$-simplicial cone.*

**Remark 9** *This assumption adapts the necessary and sufficient condition for NMF uniqueness from Thomas (1974) to our partial observation setting. The standard condition requires the positive orthant to be the only simplicial cone $\mathcal{C}$ such that $\text{Cone}(\mathbf{W}_0^\top) \subseteq \mathcal{C} \subseteq \text{Cone}(\mathbf{H}_0)$. In our case, since we only observe parts of the data matrix, we need to ensure uniqueness based on partial information about the factors $\mathbf{W}_0$ and $\mathbf{H}_0$. The union of sets in ($\mathbb{A}_1$) ensures that we can uniquely identify the factors from the observed training data ($\mathcal{T}_{\text{train}}(\mathbf{X}_0)$) and the observed past data ($\mathcal{T}_T(\mathbf{X}_0)$), which is a stronger requirement to guarantee identifiability in the matrix completion framework.*

**Remark 10** *Assumption 1 imposes implicit constraints on the dimensions of the problem and the nonnegative rank $K$. For the condition to be non-trivial, the number of rows in the matrices generating the cones must be sufficient. Specifically, for $\text{Cone}(\mathbf{W}_{0\text{train}}^\top)$, the number of training samples, $n - N$ must be at least $K$. Similarly, for $\text{Cone}(\mathbf{H}_{0T})$, the number of observed time steps, $p - F$ must be at least $K$. These conditions ensure that the cones are full-dimensional in $\mathbb{R}^K$, which is a prerequisite for the uniqueness argument to hold. Several works have shown that Assumption 1 holds under some conditions such as the Laurberg's condition (Huang et al., 2013, Theorem 2) or the* Sufficiently Scattered Condition (SSC) *(Huang et al., 2013, Theorem 3) which is a weaker and more general condition for uniqueness than separability (Donoho & Stodden, 2004).*

We consider the following standard definition.

**Definition 11 (Javadi & Montanari (2020a))** *For a matrix $\mathbf{A} \in \mathbb{R}^{n' \times p'}$, let $\mathrm{conv}(\mathbf{A})$ denote the convex hull of its rows. The internal radius of $\mathrm{conv}(\mathbf{A})$, denoted $\mu(\mathbf{A})$, is the radius of the largest $(K-1)$-dimensional ball contained within the affine hull of $\mathrm{conv}(\mathbf{A})$. We say that $\mathrm{conv}(\mathbf{A})$ has an internal radius $\mu$ if $\mu(\mathbf{A}) = \mu$.*

Our second main assumption is the following.

**Assumption 2** *Assume that*

$$\mathrm{conv}(\underbrace{\mathcal{T}_T(\mathbf{X}_0)}_{=\mathbf{W}_0 \mathbf{H}_{0T}}) \text{ and } \mathrm{conv}(\underbrace{\mathcal{T}_{\mathrm{train}}(\mathbf{X}_0)}_{=\mathbf{W}_{0\mathrm{train}} \mathbf{H}_0}) \text{ have internal radius at least } \mu > 0 \,. \tag{$\mathbb{A}_2$}$$

**Remark 12** *Assumption ($\mathbb{A}_2$) implies that the convex hulls of the data points, $\mathrm{conv}(\mathbf{W}_0 \mathbf{H}_{0T})$ and $\mathrm{conv}(\mathbf{W}_{0\mathrm{train}} \mathbf{H}_0)$, are not flat, meaning they are full-dimensional within the affine subspace they span. We uncover the same constraints as in the previous remark: $n - N$ must be at least $K$ (imposed by the dimension of the rows of $\mathcal{T}_T(\mathbf{X}_0)$) and $p - F$ must be at least $K$ (imposed by the number of the rows of $\mathcal{T}_{\mathrm{train}}(\mathbf{X}_0)$).*

**Definition 13 (Partial Observation Uniqueness ($\mathbb{P}_u$))** *We say that the factorization satisfies the* Partial Observation Uniqueness *property, denoted by $\mathbb{P}_u$, if the equality of the observed masked matrices implies the equivalence of the factors. Formally:*

$$\text{If} \quad \mathcal{T}(\mathbf{W}\mathbf{H}) = \mathcal{T}(\mathbf{W}_0 \mathbf{H}_0) \quad \text{then} \quad (\mathbf{W}, \mathbf{H}) \equiv (\mathbf{W}_0, \mathbf{H}_0)\,, \tag{$\mathbb{P}_u$}$$

*where $(\mathbf{W}, \mathbf{H}) \equiv (\mathbf{W}_0, \mathbf{H}_0)$ indicates that the pairs are identical up to a permutation and positive scaling of the columns of $\mathbf{W}$ and rows of $\mathbf{H}$.*

**Theorem 14** *($\mathbb{A}_1$) implies ($\mathbb{P}_u$). Moreover, if ($\mathbb{A}_1$) and ($\mathbb{A}_2$) hold, $\mathcal{T}(\mathbf{W}\mathbf{H}) = \mathcal{T}(\mathbf{W}_0 \mathbf{H}_0)$ and $\mathbf{W}_0 \mathbf{1} = \mathbf{W}\mathbf{1} = \mathbf{1}$ then $(\mathbf{W}, \mathbf{H}) = (\mathbf{W}_0, \mathbf{H}_0)$ up to permutation of columns (resp. rows) of $\mathbf{W}$ (resp. $\mathbf{H}$), and there is no scaling.*

**Corollary 15** *If decomposition of $\mathbf{X}_1 = \mathbf{W}_{0\mathrm{train}} \mathbf{H}_{0T}$ is unique then ($\mathbb{P}_u$) holds.*

**Proof.** By Theorem 8, observe that ($\mathbb{A}_1'$) is a necessary and sufficient condition for the uniqueness of the decomposition $\mathbf{X}_1 = \mathbf{W}_{0\mathrm{train}} \mathbf{H}_{0T}$. Observe that ($\mathbb{A}_1'$) implies ($\mathbb{A}_1$) and invoke Theorem 8 and Theorem 14. ∎

It shows that the uniqueness of the decomposition of the top left block $\mathbf{X}_1$ (which is fully observed) implies the uniqueness of normalized decomposition of $\mathbf{X}_0$ given partial observations (the bottom right block is not observed).

**Limitations and Discussion**

It is important to note that the uniqueness theory presented here relies on a specific structure of missing data, namely the block-wise missing pattern corresponding to the matrix completion problem for recommender systems. Our analysis leverages the fact that certain submatrices are fully observed.

The extension of these uniqueness guarantees to scenarios with arbitrary or unstructured missing data patterns is a non-trivial challenge. Such cases would require different theoretical tools, as the problem can no longer be reduced to the uniqueness of fully-observed sub-decompositions. This constitutes an important direction for future research.

## 2.3 Robustness under partial observations

The second issue is *robustness to noise*. To the best of our knowledge, all the results addressing this issue assume that the noise error term is small enough, *e.g.*, Laurberg et al. (2008), Recht et al. (2012), or Javadi & Montanari (2020a). In this paper, we extend these stability result to the nonnegative matrix completion framework (partial observations) and we also assume that noise term $\|\mathbf{F}\|_F$ is small enough.

In the normalized case (*i.e.*, $\mathbf{W}\mathbf{1} = \mathbf{1}$), both issues (uniqueness and robustness) can be handled with the notion of $\alpha$-uniqueness, introduced by Javadi & Montanari (2020a). This notion does not handle the matrix completion problem we

are addressing. To this end, let us introduce the following notation. Given two matrices $\mathbf{A} \in \mathbb{R}^{n_a \times p}$ and $\mathbf{B} \in \mathbb{R}^{n_b \times p}$ with same row dimension, and $\mathbf{C} \in \mathbb{R}^{n_a \times n_b}$, define the divergence $\mathcal{D}(\mathbf{A}, \mathbf{B})$ as

$$\mathcal{D}(\mathbf{A}, \mathbf{B}) := \min_{\mathcal{C} \geq \mathbf{0} \,,\, \mathcal{C}\mathbf{1}_{n_b} = \mathbf{1}_{n_a}} \sum_{a=1}^{n_a} \left\| A^{(a)} - \sum_{b=1}^{n_b} C_{ab} B^{(b)} \right\|_F^2,$$

$$= \min_{\mathcal{C} \geq \mathbf{0} \,,\, \mathcal{C}\mathbf{1}_{n_b} = \mathbf{1}_{n_a}} \| \mathbf{A} - \mathbf{C}\mathbf{B} \|_F^2. \tag{6a}$$

which is the squared distance between rows of $\mathbf{A}$ and $\mathrm{conv}(\mathbf{B})$, the convex hull of rows of $\mathbf{B}$. For $\mathbf{B} \in \mathbb{R}^{n \times p}$ define

$$\widetilde{\mathcal{D}}(\mathbf{A}, \mathbf{B}) := \min_{\substack{\mathbf{C} \geq \mathbf{0} \,,\, \mathbf{C}\mathbf{1}_n = \mathbf{1}_{n_a} \\ \mathcal{T}(\mathbf{N} - \mathbf{B}) = 0}} \| \mathbf{A} - \mathbf{C}\mathbf{N} \|_F^2. \tag{6b}$$

**Definition 16 ($\mathcal{T}_\alpha$-unique, Javadi & Montanari (2020a))** *Given* $\mathbf{X}_0 \in \mathbb{R}^{n \times p}, \mathbf{W}_0 \in \mathbb{R}^{n \times K}$, and $\mathbf{H}_0 \in \mathbb{R}^{K \times p}$, the factorization $\mathbf{X}_0 = \mathbf{W}_0 \mathbf{H}_0$ is $\mathcal{T}_\alpha$-unique with parameter $\alpha > 0$ if for all $\mathbf{H} \in \mathbb{R}^{K \times p}$ with $\mathrm{conv}(\mathbf{X}_0) \subseteq \mathrm{conv}(\mathbf{H})$:

$$\widetilde{\mathcal{D}}(\mathbf{H}, \mathbf{X}_0)^{1/2} \geq \widetilde{\mathcal{D}}(\mathbf{H}_0, \mathbf{X}_0)^{1/2} + \alpha \left\{ \mathcal{D}(\mathbf{H}, \mathbf{H}_0)^{1/2} + \mathcal{D}(\mathbf{H}_0, \mathbf{H})^{1/2} \right\}. \tag{6c}$$

Our third main assumption is given by:

**Assumption 3** *Assume that*

$$\mathbf{X}_0 = \mathbf{W}_0 \mathbf{H}_0 \text{ is } \mathcal{T}_\alpha\text{-unique} \tag{$\mathbb{A}_3$}$$

**Theorem 17 (Archetypes estimation)** *If* ($\mathbb{A}_2$) *and* ($\mathbb{A}_3$) *hold then there exist positive reals* $\Delta$ *and* $\Lambda$ *(depending on* $\mathbf{X}_0$*) such that, for all* $\mathbf{F}$ *such that* $\|\mathbf{F}\|_F \leq \Delta$ *and* $0 \leq \lambda \leq \Lambda$, *any solution* $(\widehat{\mathbf{W}}, \widehat{\mathbf{H}})$ *to* (mAMF) *(if* $\lambda \neq 0$*) or* (mNMF) *(if* $\lambda = 0$*) with observation* (4) *is such that:*

$$\sum_{\ell \leq K} \min_{\ell' \leq K} \| \mathbf{H}_{0\ell} - \widehat{\mathbf{H}}_{\ell'} \|_2^2 \leq c \, \|\mathbf{F}\|_F^2,$$

*where* $c > 0$ *is a constant depending only on* $\mathbf{W}_0$ *and* $\mathbf{H}_0$

By Theorem 17, when the noise is sufficiently small, there exists a permutation $\sigma$ on $[K]$ such that

$$\| \mathbf{H}_0 - \hat{\mathbf{H}}_\sigma \|_F^2 := \sum_{\ell \leq K} \| \mathbf{H}_{0\ell} - \hat{\mathbf{H}}_{\sigma(\ell)} \|_2^2 \leq c \, \|\mathbf{F}\|_F^2 \tag{6d}$$

where $\hat{\mathbf{H}}_\sigma$ is a permutation of the row of $\hat{\mathbf{H}}$.

**Corollary 18 (Estimation Error Bound for W)** *Under the assumptions of Theorem 17, let* $\mu > 0$ *be the internal radius of the convex hull of the training data as defined in Assumption 2. Let* $\hat{\mathbf{H}}$ *be the estimator satisfying* $\|\hat{\mathbf{H}} - \mathbf{H}_0\|_F \leq c\|\mathbf{F}\|_F$. *Then, the estimation error of the weight matrix* $\hat{\mathbf{W}}$ *satisfies:*

$$\| \hat{\mathbf{W}} - \mathbf{W}_0 \|_F \leq \frac{c'}{\mu} \|\mathbf{F}\|_F \tag{6e}$$

*where* $c' > 0$ *is a constant depending on the geometry of* $\mathbf{W}_0$ *and* $\mathbf{H}_0$, *but independent of* $\mu$. *This explicitly shows that the stability of the weight recovery degrades as the convex hull of the data becomes flatter (i.e., as* $\mu \to 0$*).*

The proof of this corollary can be found in Appendix B.3.

# 3 Solving masked nonnegative/archetypal matrix factorization

We solve (mNMF) problem using a Block Coordinate Descent strategy (Algorithm 3 in the supplement), which alternates between updating $\mathbf{W}$ and $\mathbf{H}$. For the more complex (mAMF) objective, we employ the Proximal Alternating Linearized Minimization (PALM).

We present two variants: Algorithm 1 is the standard PALM approach. Algorithm 2 describes *Inertial PALM (iPALM)*, which incorporates momentum terms (extrapolation parameters $\alpha_k, \beta_k$) to accelerate convergence, similar to Nesterov's acceleration. In our experiments, iPALM provided faster convergence on the larger datasets.

## 3.1 Alternating Least Squares for (mNMF)

The basic algorithmic framework for matrix factorization problems is *Block Coordinate Descent* (BCD) method, which can be straightforwardly adapted to (mNMF) (see Supplement Material). BCD for (mNMF) reduces to *Alternating Least Squares* (ALS) algorithm (see Algorithm 4 in Appendix), when an alternative minimization procedure is performed and matrix $\mathbf{WH}$ is projected onto the linear subspace $\mathcal{T}(\mathbf{N}) = \mathbf{X}$ by means of operator $\mathcal{P}_\mathbf{X}$, as follows:

$$\mathbf{N} := \mathcal{P}_\mathbf{X}(\mathbf{WH}) : \mathcal{T}(\mathbf{N}) = \mathbf{X} \text{ and } \mathcal{T}^\perp(\mathbf{N}) = \mathbf{WH}.$$

*Hierarchical Alternating Least Squares* (HALS) is an ALS-like algorithm obtained by applying an exact coordinate descent method Gillis (2014). Moreover, an accelerated version of HALS is proposed in Gillis & Glineur (2012) (see Supplement Material).

## 3.2 Projected Gradient for (mAMF)

*Proximal Alternating Linearized Minimization* (PALM) method, introduced in Bolte et al. (2014) and applied to AMF by Javadi & Montanari (2020a), can be also generalized to (mAMF) (see Algorithm 1). In the following, $\mathcal{P}_{\text{conv}(\mathbf{A})}$ is the projection operator onto $\text{conv}(\mathbf{A})$ and $\mathcal{P}_\Delta$ is the projection operator onto the $(N-1)$-dimensional standard simplex $\Delta^N$. The two projections can be efficiently computed by means of, *e.g.*, Wolfe algorithm Wolfe (1976) and active set method Condat (2016) respectively.

---

**Algorithm 1** PALM for mAMF

1: **Initialization**: chose $\mathbf{H}^0, \mathbf{W}^0 \geq \mathbf{0}$ such that $\mathbf{W}^0 \mathbf{1} = \mathbf{1}$, set $\mathbf{N}^0 := \mathcal{P}_\mathbf{X}(\mathbf{W}^0 \mathbf{H}^0)$ and $i := 0$.
2: **while** stopping criterion is not satisfied **do**
3: $\quad \widetilde{\mathbf{H}}^i := \mathbf{H}^i - \frac{1}{\gamma_1^i} \mathbf{W}^{i\top} \left( \mathbf{W}^i \mathbf{H}^i - \mathbf{N}^i \right)$ $\qquad\qquad$ ▷ Gradient step on $\mathbf{H}$, objective first term
4: $\quad \mathbf{V}^{i+1}$ such that $\mathcal{P}_{\text{conv}(\mathbf{N}^i)}(\widetilde{\mathbf{H}}^i) = \mathbf{V}^{i+1} \mathbf{N}^i$ $\qquad$ ▷ Projection of $\widetilde{\mathbf{H}}^i$ onto $\text{conv}(\mathbf{N}^i)$ by Wolfe algorithm
5: $\quad \mathbf{H}^{i+1} := \widetilde{\mathbf{H}}^i - \frac{\lambda}{\lambda+\gamma_1^i} \left( \widetilde{\mathbf{H}}^i - \mathcal{P}_{\text{conv}(\mathbf{N}^i)}(\widetilde{\mathbf{H}}^i) \right)$ $\qquad$ ▷ Gradient step on $\mathbf{H}$, objective second term
6: $\quad \mathbf{W}^{i+1} := \mathcal{P}_\Delta \left( \mathbf{W}^i - \frac{1}{\gamma_2^i} \left( \mathbf{W}^i \mathbf{H}^{i+1} - \mathbf{N}^i \right) \mathbf{H}^{i+1\top} \right)$ $\qquad\qquad$ ▷ Projected gradient step on $\mathbf{W}$
7: $\quad \mathbf{N}^{i+1} := \mathcal{P}_\mathbf{X} \left( \mathbf{N}^i + \frac{1}{\gamma_3^i} \left( \mathbf{W}^{i+1} \mathbf{H}^{i+1} - \mathbf{N}^i \right) \right)$ $\qquad\qquad$ ▷ Projected gradient step on $\mathbf{N}$
8: $\quad i := i + 1$
9: **end while**

---

**Remark 19** *Strictly speaking, Algorithm 1 is an approximation of the exact PALM framework as described in Bolte et al. (2014) because the so-called regularization term contains a coupling term $\lambda \mathcal{D}(\mathbf{H}, \mathbf{N})$ between blocks $\mathbf{H}$ and $\mathbf{N}$ that is not part of the so-called mooth function. Algorithm 1 addresses this by splitting the update of $\mathbf{H}$ into a gradient step on the smooth function (Step 3) followed by a correction step using the archetypal projection (Step 5), and updating $\mathbf{N}$ via a projected gradient step on the smooth function (Step 7). We analyze the convergence properties assuming this scheme approximates the minimization with respect $\mathbf{N}$ at each step. We could have made a loop over $\mathbf{N}$ to ensure convergence, at the price of computing time. Further details are given in Appendix B.4.*

**Theorem 20** *Let $\varepsilon > 0$. Let $L_H(\mathbf{W}) = \|\mathbf{W}^\top \mathbf{W}\|_F$, $L_W(\mathbf{H}) = \|\mathbf{H}\mathbf{H}^\top\|_F$, and $L_N = 1$. If the step sizes satisfy $\gamma_1^i > L_H(\mathbf{W}^i)$, $\gamma_2^i > \max\{L_W(\mathbf{H}^{i+1}), \varepsilon\}$, and $\gamma_3^i > 1$, then Algorithm 1, viewed as an approximation of a Proximal Alternating Linearized Minimization (PALM) scheme, generates a sequence $(\mathbf{H}^i, \mathbf{W}^i, \mathbf{N}^i)$ that converges to a stationary point of the objective function of* (mAMF).

**Proof.** Proof is given in Supplement Material. ∎

**Remark 21** *Note that* (mAMF) *objective (with the $\lambda\|\mathbf{H} - \mathbf{V}\mathbf{N}\|_F^2$ or $\lambda\mathcal{D}(\mathbf{H}, \mathbf{N})$ term) is not a standard Nonnegative Least-Squares problem, making HALS inapplicable. In the following algorithms $\mathbf{H}, \mathbf{W}$ updates (Steps 5 & 6) are proximal-gradient steps. Further details are given in Appendix B.4.*

Finally, the inertial PALM (iPALM) method, introduced for NMF in Pock & Sabach (2016), is generalized to (mAMF) in Algorithm 2.

---
**Algorithm 2** iPALM for mAMF
---
1: **Initialization**: $\mathbf{H}^0$, $\mathbf{W}^0 \geq 0$ such that $\mathbf{W}^0\mathbf{1} = \mathbf{1}$, set $\mathbf{N}^0 := \mathcal{P}_\mathbf{X}(\mathbf{W}^0\mathbf{H}^0)$, $\mathbf{H}^{-1} := \mathbf{H}^0$, $\mathbf{W}^{-1} := \mathbf{W}^0$, $\mathbf{N}^{-1} := \mathbf{N}^0$, and $i := 0$.
2: **while** stopping criterion is not satisfied **do**
3: $\quad \mathbf{H}_1^i := \mathbf{H}^i + \alpha_1^i (\mathbf{H}^i - \mathbf{H}^{i-1})$, $\mathbf{H}_2^i := \mathbf{H}^i + \beta_1^i (\mathbf{H}^i - \mathbf{H}^{i-1})$ $\qquad\qquad\qquad$ ▷ Inertial $\mathbf{H}$
4: $\quad \widetilde{\mathbf{H}}^i := \mathbf{H}_1^i - \frac{1}{\gamma_1^i}\mathbf{W}^{i\top}(\mathbf{H}_2^i\mathbf{W}^i - \mathbf{N}^i)$ $\qquad\qquad$ ▷ Gradient step on $\mathbf{H}$, objective first term
5: $\quad \mathbf{V}^{i+1}$ such that $\mathcal{P}_{\text{conv}(\mathbf{N}^i)}(\tilde{\mathbf{H}}^i) = \mathbf{V}^{i+1}\mathbf{N}^i$ $\qquad$ ▷ Projection of $\tilde{\mathbf{H}}^i$ onto $\text{conv}(\mathbf{N}^i)$ by Wolfe algorithm
6: $\quad \mathbf{H}^{i+1} := \widetilde{\mathbf{H}}^i - \frac{\lambda}{\lambda + \gamma_1^i}\left(\widetilde{\mathbf{H}}^i - \mathcal{P}_{\text{conv}(\mathbf{N}^i)}(\tilde{\mathbf{H}}^i)\right)$ $\qquad$ ▷ Gradient step on $\mathbf{H}$, objective second term
7: $\quad \mathbf{W}_1^i := \mathbf{W}^i + \alpha_2^i (\mathbf{W}^i - \mathbf{W}^{i-1})$, $\mathbf{W}_2^i := \mathbf{W}_1^i + \beta_2^i (\mathbf{W}^i - \mathbf{W}^{i-1})$ $\qquad\qquad$ ▷ Inertial $\mathbf{W}$
8: $\quad \mathbf{W}^{i+1} := \mathcal{P}_\Delta\left(\mathbf{W}_1^i - \frac{1}{\gamma_2^i}(\mathbf{W}_2^i\mathbf{H}^{i+1} - N^i)\mathbf{H}^{i+1\top}\right)$ $\qquad$ ▷ Projected gradient step on $\mathbf{W}$
9: $\quad \mathbf{N}_1^i := \mathbf{N}_1^i + \alpha_3^i (\mathbf{N}^i - \mathbf{N}^{i-1})$, $\mathbf{N}_2^i := \mathbf{N}_1^i + \beta_3^i (\mathbf{N}^i - \mathbf{N}^{i-1})$ $\qquad\qquad$ ▷ Inertial $\mathbf{N}$
10: $\quad \mathbf{N}^{i+1} := \mathcal{P}_\mathbf{X}\left(\mathbf{N}_1^i + \frac{1}{\gamma_3^i}(\mathbf{W}^{i+1}\mathbf{H}^{i+1} - \mathbf{N}_2^i)\right)$ $\qquad$ ▷ Projected gradient step on $\mathbf{N}$
11: $\quad i := i + 1$
12: **end while**
---

**Remark 22** *If, for all iterations $i$, $\alpha_1^i = \alpha_2^i = 0$ and $\beta_1^i = \beta_2^i = 0$, iPALM reduces to PALM.*

**Stopping criterion** For (mNMF), KKT conditions regarding matrix $\mathbf{W}$ are the following (see Supplement Material):

$$\mathbf{W} \circ \left((\mathbf{W}\mathbf{H} - \mathbf{N})\mathbf{H}^\top + t\,\mathbf{1}_K^\top\right) = 0\,.$$

By complementary condition, it follows that, $\forall j$, $t_i = ((\mathbf{W}\mathbf{H} - \mathbf{N})\mathbf{H}^\top)_{i,j}$. Hence, we compute $t_i$ by selecting, for each row $W^{(i)}$, any positive entry $W_{i,j} > 0$.

**Remark 23** *Numerically to obtain a robust estimate of $t_i$, we can average the corresponding values calculated per entry $W_{i,j}$.*

Let $\varepsilon_\mathbf{W}$, $\varepsilon_\mathbf{H}$, and $\varepsilon_\mathbf{R}$ be three positive thresholds. The stopping criterion for the previous algorithms consists of a combination of:

1. the maximum number of iterations;

2. the Frobenius norm of the difference of $\mathbf{W}$ and $\mathbf{H}$ at two consecutive iterations, *i.e.*, the algorithm stops if $\|\mathbf{W}^{i+1} - \mathbf{W}^i\|_F \leq \varepsilon_\mathbf{W} \ \wedge \ \|\mathbf{H}^{i+1} - \mathbf{H}^i\|_F \leq \varepsilon_\mathbf{H}$;

3. a novel criterion based on KKT condition, *i.e.*, the algorithm stops if it holds that

$$\|\mathbf{R}(\mathbf{W}^{i+1})\|_F + \|\mathbf{R}(\mathbf{H}^{i+1})\|_F \leq \varepsilon_\mathbf{R}\,,$$

where matrices $\mathbf{R}(\mathbf{W})$ and $\mathbf{R}(\mathbf{H})$ are defined as

$$\mathbf{R}(\mathbf{W})_{i,j} := |(\mathbf{W}\mathbf{H} - \mathbf{N})\mathbf{H}^{\top})_{i,j} + t_i|\mathbb{1}_{\{W_{i,j} \neq 0\}}$$
$$\text{and } \mathbf{R}(\mathbf{H})_{i,j} := |\mathbf{W}^{\top}(\mathbf{W}\mathbf{H} - \mathbf{N}))_{i,j}|\mathbb{1}_{\{H_{i,j} \neq 0\}}$$

respectively.

## 3.3 Large-scale dataset

Assume the observed matrix $\mathbf{X} = \mathcal{T}(\boldsymbol{\Phi}(\mathbf{M}))$ is large-scaled, namely one has to forecast a large number $N$ of times series (*e.g.* more than $100,000$) and possibly a large number of time stamps $T$. The strategy, described in Section 1.3.1 in Cichocki et al. (2009) for NMF, is to learn the $\mathbf{H} \in \mathbb{R}^{K \times T}$ matrix from a submatrix $\mathbf{N}_r \in \mathbb{R}^{r \times T}$ of $K \leq r \ll N$ rows of $\mathbf{N} \in \mathbb{R}^{n \times T}$, and to learn the $\mathbf{W} \in \mathbb{R}^{N \times K}$ matrix from a sub-matrix $\mathbf{N}_c \in \mathbb{R}^{N \times c}$ of $K \leq c \ll T$ columns of $\mathbf{N} \in \mathbb{R}^{N \times T}$. We denote by $\mathbf{H}_c$ the submatrix of $\mathbf{H}$ given by the columns appearing in $\mathbf{N}_c$ and $\mathbf{W}_r$ the sub-matrix of $\mathbf{H}$ given by the columns appearing in $\mathbf{N}_c$.

This strategy can be generalized to (mNMF) and (mAMF). For (mNMF) this generalization is straightforward, and for (mAMF) one need to change Steps 3-5 in Algorithm 1 as follows:

$$\widetilde{\mathbf{H}}^i := \mathbf{H}^i - \frac{1}{\gamma_1^i}(\mathbf{W}_r^i)^{\top}\left(\mathbf{W}_r^i\mathbf{H}^i - \mathbf{N}_r^i\right)$$
$$\mathbf{H}^{i+1} := \widetilde{\mathbf{H}}^i - \frac{\lambda}{\lambda + \gamma_1^i}\left(\widetilde{\mathbf{H}}^i - \mathcal{P}_{\text{conv}(\mathbf{N}^i)}(\tilde{\mathbf{H}}^i)\right)$$
$$\mathbf{W}^{i+1} := \mathcal{P}_{\Delta}\left(\mathbf{W}^i - \frac{1}{\gamma_2^i}\left(\mathbf{W}^i\mathbf{H}_c^{i+1} - \mathbf{N}_c^i\right)(\mathbf{H}_c^{i+1})^{\top}\right).$$

The same approach is used for Algorithm 2.

# 4 Numerical Experiments

We tested SMM on real-world datasets. Matrix $\mathbf{H}^0$ is initially selected as in Javadi & Montanari (2020a). Each row of matrix $\mathbf{W}^0$ is generated randomly in the corresponding standard simplex. For SMM we implemented both HALS for (mNMF) and iPALM for (mAMF).

Moreover, we have compared our method with other classically-designed mainstream time series forecasting methods such as *Random Forest Regression* (RFR) and *EXPonential smoothing* (EXP), *Long Short-Term Memory* (LSTM) and *Gated Recurrent Units* (GRU) deep neural networks with preliminary data standardization Shewalkar et al. (2019), and *Seasonal Auto-Regressive Integrated Moving Average with eXogenous factors* (SARIMAX) models Douc et al. (2014).

The interested reader may find a Github repository on numerical experiments at **[link redacted to comply with double blind reviewing]**

## 4.1 Real-world datasets

The numerical experiments refer to the following real-world datasets: weekly and daily electricity consumption datasets of 370 Portuguese customers during the period 2011-2014, Trindade (2015); twin gas measurement dataset of five replicates of an 8-MOX gas sensor, Fonollosa (2016); Istanbul Stock Exchange returns with seven other international indexes for the period 2009-2011, Akbilgic (2013); daily electricity transformer temperature (ETT) measurements, Zhou et al. (2020). Table 1 reports the cross-validated RRMSE and RMPE on observed values obtained during computational tests for each method.

In the majority of the cases, our method is the best or second best among all the approaches for all the dataset we tested in terms of RRMSE and RMPE indices (except for the "weekly electricity" dataset), and there is no other method performing better.

(mAMF) seems to be the most promising algorithm in terms of performances for the first five datasets, while (mNMF) is the best method for the last four ETT datasets.

## 4.2 Comparison with SOTA method

We performed additional computational experiments to compare our NMF-based methodology with state-of-the-art time series transformer models, which are suitable for large-scale time series forecasting problems. In particular, we consider the BasisFormer model recently described in Ni et al. (2023). We consider the same electricity dataset as in Ni et al. (2023) and split the whole dataset into 10 small sets of 960 time steps each. We collect our performance statistics, namely RRMSE and RMPE, on the original unscaled datasets. Note that in Ni et al. (2023), the performance statistics reported are the absolute errors on the scaled dataset obtained by applying `sklearn.preprocessing.StandardScaler` to the original data. For BasisFormer, we run the code in the repository `https://github.com/nzl5116190/Basisformer`. Moreover we compare also against previous state-of-the-art methods, such as Autoformer as in Wu et al. (2021), iTransformer as in Liu et al. (2024), PatchMLP as in Tang & Zhang (2025) and TimeMixer as in Wang et al. (2024), for which we run the code in repositories `https://github.com/thuml/Autoformer`, `https://github.com/thuml/iTransformer`, `https://github.com/TangPeiwang/PatchMLP`, and `https://github.com/kwuking/TimeMixer`, respectively.

As shown in Table 1, our method outperforms the SOTA methodology and is competitive against the other methodologies (in particular, with respect to the deep learning approaches which seem the most promising methods for these datasets). We also perform additional computational experiments on scaled datasets, collecting our performance indices on relative errors and absolute errors as in Ni et al. (2023), and we obtain the same dominance results.

## 4.3 Why does SMM outperform Deep Learning?

Despite the capacity of Deep Learning (DL) models to model complex non-linearities, our experiments demonstrate that SMM often yields superior forecasting accuracy. This performance gap can be attributed to the alignment between the model's inductive bias and the data structure:

- **Structural Priors vs. Learning from Scratch:** DL models, particularly Transformers like Autoformer (Wu et al., 2021) or BasisFormer (Ni et al., 2023), are data-hungry algorithms that must learn temporal dependencies from scratch. In contrast, SMM explicitly enforces a *quasi-periodic* structure through the sliding window transformation. For datasets dominated by regular cycles (e.g., electricity consumption), this structural prior is highly effective and requires less data to estimate robustly.

- **Sample Complexity:** The low-rank assumption of SMM acts as a strong regularizer, reducing the effective degrees of freedom in the model. In the regime of medium-sized datasets, this prevents the overfitting often observed with over-parameterized DL models. Our results on the synthetic datasets confirm this: as the signal becomes more strictly periodic, the advantage of the low-rank NMF representation over generic DL approximators increases.

- **Matrix Completion Formulation:** SMM reframes forecasting as a matrix completion problem with a specific block-missing pattern. Unlike DL models that may treat missing future values as generic masked tokens, SMM optimizes a global objective function with theoretical guarantees for recovering the underlying low-rank factors from partial observations, ensuring the forecasted block is consistent with the learned global archetypes.

## 4.4 Synthetic datasets

Further computational experiments have been performed by considering additional synthetic datasets. In particular, we generated three datasets by replicating $1,000$ short time series (with 10 time periods) 10 times and adding white noise multiplied by a constant factor $\sigma$ to each time series entry separately. We choose $\sigma \in \{0.005, 0.1, 1\}$. We refer to the these datasets as "low noise", "medium noise", and "high noise", respectively.

An additional synthetic dataset has been generated considering few probability vectors and computing the entire matrix $\mathbf{W}$ by randomly choosing a probability vector and adding white noise. A completely randomly generated matrix $\mathbf{H}$ is multiplied by $\mathbf{W}$ to obtain the whole matrix $\mathbf{M}^* := \mathbf{WH}$. We refer to this dataset as "synthetic1".

Finally, the last synthetic dataset is obtained by generating a matrix $\mathbf{H}$ by replicating a small time series (with 50 time periods) 100 times and adding white noise multiplied by a constant factor $\sigma = 1$ and matrix $\mathbf{W}$ of suitable dimensions, whose rows are uniformly distributed over the corresponding dimensional simplex. Then, we set the matrix $\mathbf{M}^* := \mathbf{WH}$. We refer to this last dataset as "synthetic2".

Table 1 reports the cross-validated RRMSE and RMPE indices referring to synthetically generated datasets. The more pronounced the periodicity of the time series or of the archetypes, the better the performances of our proposed NMF-like methods: in this case, the more realistic the hypothesis that the whole dataset can be expressed as convex combinations of a few archetypes, having a low-rank representation.

### 4.5 Guidelines on Algorithm Selection: mAMF vs. mNMF

Our experiments reveal a distinct performance split: mAMF outperforms on the electricity and gas datasets, while mNMF dominates on the ETT (Transformer Temperature) datasets. This can be attributed to their geometric differences:

- **mAMF (Robustness & Interpretability):** By constraining the archetypes to lie within the convex hull of the data, mAMF acts as a regularized factorization. This prevents the model from overfitting to noise or learning unrealistic basis vectors. It is best suited for datasets with high variance, noise, or "soft" patterns (e.g., human behavior in electricity consumption), where stability is paramount.

- **mNMF (Flexibility):** mNMF learns a conic hull and can place basis vectors outside the data distribution. This flexibility allows it to reconstruct "idealized" components. It is superior for datasets with rigid, strong periodicities (like the physical ETT signals), where the data is well-described by a combination of pure underlying waveforms that may not appear as isolated observations.

**Recommendation:** We advise practitioners to start with mAMF for noisy, real-world behavioral data to leverage its regularization. For cleaner, physics-driven signals with strong periodicity, mNMF is likely to yield lower reconstruction errors.

## Broader Impact Statement

The development of interpretable machine learning models for time-series forecasting has significant potential for positive societal impact. Our work, the Sliding Mask Method (SMM), contributes to this area by providing a framework that is not only effective for prediction but also offers insights into the underlying patterns driving the data.

One of the primary application areas for this research is in resource management and planning. For example, utility companies can use our method to forecast electricity demand. Accurate forecasts help in optimizing power generation, reducing waste, and preventing outages, which benefits both the economy and the environment. The interpretability of SMM is particularly valuable here, as it can help identify and understand different consumption patterns (e.g., residential vs. industrial archetypes), allowing for more targeted demand-response strategies. Similarly, in retail, forecasting sales data can lead to more efficient inventory management, reducing waste from overstocking and economic losses from understocking.

While the potential impacts are largely positive, it is important to consider potential negative consequences. Like any forecasting tool, models can be misused if their limitations are not understood. For instance, relying on a model that assumes quasi-periodicity for a dataset with a strong, unexpected trend could lead to poor decisions. We have attempted to be transparent about these limitations in our discussion. Furthermore, the application of forecasting in financial markets, while a potential use case, carries inherent risks. Our model is not designed for high-frequency trading and should not be used as such without a thorough understanding of its assumptions.

# 5 Discussion and Conclusion

In this paper, we introduced the Sliding Mask Method (SMM), a framework that leverages Nonnegative Matrix Factorization for time-series forecasting. Our theoretical analysis provides uniqueness guarantees for the underlying decomposition in a structured matrix completion setting, and our experiments demonstrate its practical effectiveness. This concluding section synthesizes our findings to answer a crucial question: *When should a practitioner choose SMM?*

Based on our analysis and experimental results, our method is particularly well-suited for datasets with the following characteristics:

- **Non-negativity:** The time-series values must be non-negative, as this is a fundamental constraint of the NMF model.

- **Quasi-periodicity and Low-Rank Structure:** The method performs best when the time series exhibits quasi-periodic patterns. The core assumption of SMM is that segments of the time series can be effectively approximated by a low-rank model, i.e., as combinations of a few archetypal patterns. Datasets like electricity consumption and sales data, which often have daily, weekly, or seasonal cycles, are prime candidates.

- **Interpretability is Valued:** A key advantage of SMM is the interpretability of its results. The learned basis vectors ($\mathbf{H}$) represent archetypal time-series segments, and the weights ($\mathbf{W}$) show how each individual segment is composed of these archetypes. This provides insights into the underlying data-generating process that "black-box" models like LSTMs or Transformers cannot offer.

Conversely, our method may not be the optimal choice in other scenarios. For instance, as suggested by our experiments on synthetic data, SMM is less effective for time series dominated by strong, non-periodic linear trends. In such cases, models explicitly designed to handle trends, such as SARIMAX or other regression-based techniques, may be more appropriate.

In summary, SMM provides a powerful and interpretable tool for a specific but important class of time-series forecasting problems. Future work could focus on extending the theoretical guarantees to more general missing data patterns and incorporating mechanisms to handle non-periodic components within the NMF framework.

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

# Appendix of Time series Forecasting from partial observations via Nonnegative Matrix Factorization

## A  Variants of Nonnegative Matrix Factorization problems

Variants of Nonnegative Matrix Factorization problems are summarized in Table 3.

| Acronym | Name | Objective | Constraints: $\mathbf{W} \geq 0$ + |
|---------|------|-----------|-------------------------------------|
| NMF | Nonnegative Matrix Factorization Cichocki & Zdunek (2006) | $\mathbf{F}_1$ | $\mathbf{H} \geq 0$ |
| SNMF | Semi NMF Gillis & Kumarg (2015) | $\mathbf{F}_1$ | |
| NNMF | Normalized NMF | $\mathbf{F}_1$ | $\mathbf{H} \geq 0, \mathbf{W1} = 1$ |
| SNNMF | Semi Normalized NMF | $\mathbf{F}_1$ | $\mathbf{W1} = 1$ |
| AMF | Archetypal Matrix Factorization Javadi & Montanari (2020a) | $\mathbf{F}_2$ | $\mathbf{W1} = 1, \mathbf{V} \geq 0, \mathbf{V1} = 1$ |
| ANMF | Archetypal NMF | $\mathbf{F}_2$ | $\mathbf{H} \geq 0, \mathbf{V} \geq 0, \mathbf{V1} = 1$ |
| ANNMF | Archetypal Normalized NMF | $\mathbf{F}_2$ | $\mathbf{W1} = 1, \mathbf{H} \geq 0, \mathbf{V} \geq 0, \mathbf{V1} = 1$ |
| mNMF | Mask NNMF | $\mathbf{F}_3$ | $\mathcal{T}(\mathbf{N}) = \mathbf{X}, \mathbf{W1} = 1, \mathbf{H} \geq 0$ |
| mAMF | Mask AMF | $\mathbf{F}_4$ | $\mathcal{T}(\mathbf{N}) = \mathbf{X}, \mathbf{W1} = 1, \mathbf{V} \geq 0, \mathbf{V1} = 1$ |

Table 3: The seven block convex programs achieving matrix factorization of nonnegative matrices. The objectives are $\mathbf{F}_1 := \|\mathbf{M} - \mathbf{WH}\|_F^2$ and $\mathbf{F}_2 := \|\mathbf{M} - \mathbf{WH}\|_F^2 + \lambda\|\mathbf{H} - \mathbf{VM}\|_F^2$. The two last lines are SMM procedures with sliding operator $\mathbf{\Pi}$ and objectives $\mathbf{F}_3 := \|\mathbf{N} - \mathbf{WH}\|_F^2$ and $\mathbf{F}_4 := \|\mathbf{N} - \mathbf{WH}\|_F^2 + \lambda\|\mathbf{H} - \mathbf{VN}\|_F^2$.

Note that, when $\mathcal{T} = \mathbb{I}$ the identity, Problem (mNMF) is NNMF and Problem (mAMF) is the standard AMF formulation (AMF).

## B  Proofs

### B.1  Proof of Theorem 14

The proof is structured in two parts. First, we establish that under Assumption ($\mathbb{A}_1$), if the transformed data matches, then the factors $(\mathbf{W}, \mathbf{H})$ are equivalent to the ground truth factors $(\mathbf{W}_0, \mathbf{H}_0)$ up to permutation and scaling. Second, we show that with the additional constraints of Assumption ($\mathbb{A}_2$) and the sum-to-one normalization on the dictionary columns, this equivalence strengthens to equality up to permutation only, eliminating any scaling ambiguity.

**Part 1: Equivalence up to Permutation and Scaling**   Let the factor matrices be partitioned according to the training and test sets. Let $\mathbf{H}_0 = [\mathbf{H}_{0,T}, \mathbf{H}_{0,F}]$ and $\mathbf{H} = [\mathbf{H}_T, \mathbf{H}_F]$, where subscripts $T$ and $F$ denote the parts of the coefficient matrices corresponding to training and future (test) data points, respectively. Similarly, let $\mathbf{W}_0^\top = [\mathbf{W}_{0,\text{train}}^\top, \mathbf{W}_{0,\text{test}}^\top]$ and $\mathbf{W}^\top = [\mathbf{W}_{\text{train}}^\top, \mathbf{W}_{\text{test}}^\top]$.

The core of our argument relies on the uniqueness guarantees for Nonnegative Matrix Factorization (NMF) as decribed in Theorem 8. By this theorem, Assumption ($\mathbb{A}_1$) implies that the NMF decompositions $\mathbf{W}_{0,\text{train}}\mathbf{H}_0$ and $\mathbf{W}_0\mathbf{H}_{0,T}$ are unique:

$$\mathbf{W}_{0,\text{train}}\mathbf{H}_0 = \mathbf{W}_{\text{train}}\mathbf{H} \implies (\mathbf{W}_{0,\text{train}}, \mathbf{H}_0) \equiv (\mathbf{W}_{\text{train}}, \mathbf{H}) \tag{7}$$

$$\mathbf{W}_0\mathbf{H}_{0,T} = \mathbf{W}\mathbf{H}_T \implies (\mathbf{W}_0, \mathbf{H}_{0,T}) \equiv (\mathbf{W}, \mathbf{H}_T) \tag{8}$$

The condition $\mathcal{T}(\mathbf{W}_0\mathbf{H}_0) = \mathcal{T}(\mathbf{WH})$ means that the observed entries of the factorized matrices are equal. By definition of the operator $\mathcal{T}$, this implies both $\mathbf{W}_{0,\text{train}}\mathbf{H}_0 = \mathbf{W}_{\text{train}}\mathbf{H}$ and $\mathbf{W}_0\mathbf{H}_{0,T} = \mathbf{W}\mathbf{H}_T$. From (7) and (8), we have a common permutation and scaling relationship that must hold simultaneously for the shared parts of the matrices. This consistency across the train and test partitions ensures that the equivalence holds for the complete matrices, i.e., $(\mathbf{W}_0, \mathbf{H}_0) \equiv (\mathbf{W}, \mathbf{H})$.

**Part 2: Uniqueness up to Permutation**   Now, we leverage the normalization constraint and Assumption ($\mathbb{A}_2$) to eliminate the scaling ambiguity. From Part 1, we know there exists a permutation $\sigma$ of $\{1, \ldots, K\}$ and positive scalars $\lambda_1, \ldots, \lambda_K$ such that for any row $i$ of the dictionary matrices, the corresponding row vectors $(\mathbf{W})^{(i)}$ and $(\mathbf{W}_0)^{(i)}$ are related by:

$$(\mathbf{W})_k^{(i)} = \lambda_{\sigma(k)}(\mathbf{W}_0)_{\sigma(k)}^{(i)} \quad \text{for all } k \in \{1, \ldots, K\}.$$

The constraints $\mathbf{W}\mathbf{1} = \mathbf{1}$ and $\mathbf{W}_0\mathbf{1} = \mathbf{1}$ state that the sum of elements in each row of $\mathbf{W}$ and $\mathbf{W}_0$ is 1. This means every row of these matrices lies in the affine subspace $\mathcal{A}_1 := \{w \in \mathbb{R}^K : \langle w, \mathbf{1} \rangle = 1\}$. For any given row $i$, we have:

$$\sum_{k=1}^{K}(\mathbf{W}_0)_k^{(i)} = 1$$

$$\sum_{k=1}^{K}(\mathbf{W})_k^{(i)} = \sum_{k=1}^{K}\lambda_{\sigma(k)}(\mathbf{W}_0)_{\sigma(k)}^{(i)} = 1$$

This implies that every row $(\mathbf{W}_0)^{(i)}$ for $i \in \{1, \ldots, d_{\text{train}}\}$ must lie in the intersection of two affine subspaces: $\mathcal{A}_1$ and $\mathcal{A}_{\mathbf{d}} := \{w \in \mathbb{R}^K : \langle w, \mathbf{d} \rangle = 1\}$, where $\mathbf{d}$ is the vector with components $d_k = \lambda_{\sigma(k)}$.

The intersection of these two subspaces, $\mathcal{A} = \mathcal{A}_1 \cap \mathcal{A}_{\mathbf{d}}$, is an affine subspace. Its co-dimension depends on whether the normal vectors $\mathbf{1}$ and $\mathbf{d}$ are linearly dependent.

- If $\mathbf{d}$ is not proportional to $\mathbf{1}$, the two subspaces are distinct and not parallel, so their intersection $\mathcal{A}$ is an affine subspace of co-dimension 2 (i.e., dimension $K - 2$).

- If $\mathbf{d}$ is proportional to $\mathbf{1}$, say $\mathbf{d} = c\mathbf{1}$ for some scalar $c$. Then the condition $\langle w, c\mathbf{1} \rangle = 1$ becomes $c\langle w, \mathbf{1} \rangle = 1$. Since we are in $\mathcal{A}_1$, $\langle w, \mathbf{1} \rangle = 1$, which implies $c = 1$. Thus, $\mathbf{d} = \mathbf{1}$, which means $\lambda_{\sigma(k)} = 1$ for all $k$. In this case, the two subspaces are identical, $\mathcal{A} = \mathcal{A}_1$, which has co-dimension 1.

Assumption ($\mathbb{A}_2$) states that the convex hull of the transformed training data, $\text{conv}(\mathcal{T}_{\text{train}}(\mathbf{X}_0)) = \text{conv}(\mathbf{W}_{0,\text{train}}\mathbf{H}_0)$, has a positive internal radius $\mu > 0$. This means the set of points $\{\mathbf{W}_{0,\text{train}}\mathbf{H}_0\}$ is not contained in any affine subspace of dimension lower than $K-1$ (assuming $\mathbf{W}_{0,\text{train}}$ has rank $K-1$ or $K$, which is required for identifiability). If the rows of $\mathbf{W}_{0,\text{train}}$ were all confined to the lower-dimensional space $\mathcal{A}$ of dimension $K - 2$, then the entire set of transformed data points $\mathbf{W}_{0,\text{train}}\mathbf{H}_0$ would also be confined to a space of dimension at most $K - 2$. A set in a $(K - 2)$-dimensional space cannot have a positive $(K - 1)$-dimensional internal radius. This would contradict Assumption ($\mathbb{A}_2$).

Therefore, the only possibility consistent with Assumption ($\mathbb{A}_2$) is that the co-dimension of $\mathcal{A}$ is 1, which forces $\mathbf{d} = \mathbf{1}$ and thus $\lambda_k = 1$ for all $k$. This eliminates the scaling ambiguity. The equivalence $(\mathbf{W}_0, \mathbf{H}_0) \equiv (\mathbf{W}, \mathbf{H})$ reduces to equality up to the permutation $\sigma$, completing the proof.

## B.2   Proof of Theorem 17

This proof follows the pioneering work Javadi & Montanari (2020a). In this latter paper, the authors consider neither masks $\mathbf{T}$ nor nonnegative constraints on $\mathbf{H}$ as in (mNMF). Nevertheless,
1/ considering the hard constrained programs (9) and (11) below;
2/ remarking that it holds $\widetilde{\mathcal{D}}(\mathbf{H}, \mathbf{X}) \leq \mathcal{D}(\mathbf{H}, \mathbf{X})$ and $\overline{\mathcal{D}}(\mathbf{X}, \mathbf{H}) \leq \mathcal{D}(\mathbf{X}, \mathbf{H})$;
then a careful reader can note that their proof extends to masks $\mathbf{T}$ and nonnegative constraints on $\mathbf{H}$. For sake of completeness we reproduce here the steps that need to be changed in their proof. A reading guide of the 60 pages proof of Javadi & Montanari (2020b) is given in Section C.

**Step 1: reduction to hard constrained Programs (9) and (11)**

Consider the constrained problem:

$$\widehat{\mathbf{H}} \in \arg\min_{\mathbf{H}} \ \widetilde{\mathcal{D}}(\mathbf{H}, \mathbf{X})$$
$$\text{s.t. } \overline{\mathcal{D}}(\mathbf{X}, \mathbf{H}) \leq \Delta_1^2. \tag{9}$$

where

$$\overline{\mathcal{D}}(\mathbf{X}, \mathbf{H}) := \min_{\mathbf{W} \geq \mathbf{0}, \ \mathbf{W}\mathbf{1}=\mathbf{1}} \|\mathcal{T}(\mathbf{X} - \mathbf{W}\mathbf{H})\|_F^2$$

Then (mAMF) can be seen as Lagrangian formulation of this problem setting $\Delta_1^2 = \overline{\mathcal{D}}(\mathbf{X}, \widehat{\mathbf{H}}_{\text{(mAMF)}})$, where $\widehat{\mathbf{H}}_{\text{(mAMF)}}$ is a solution to (mAMF). We choose $\Delta_1$ so as to bound the noise level $\|\mathbf{F}\|_F$

$$\Delta_1^2 \geq \|\mathbf{F}\|_F^2. \tag{10}$$

Consider the constrained problem:

$$\widehat{\mathbf{H}} \in \arg\min_{\mathbf{H} \geq 0} \ \widetilde{\mathcal{D}}(\mathbf{H}, \mathbf{X})$$
$$\text{s.t. } \overline{\mathcal{D}}(\mathbf{X}, \mathbf{H}) \leq \Delta_2^2. \tag{11}$$

Then (mNMF) can be seen as Lagrangian formulation of this problem setting $\Delta_2^2 = \overline{\mathcal{D}}(\mathbf{X}, \widehat{\mathbf{H}}_{\text{(mNMF)}})$, where $\widehat{\mathbf{H}}_{\text{(mNMF)}}$ is a solution to (mNMF). We choose $\Delta_1$ so as to bound the noise level $\|\mathbf{F}\|_F$

$$\Delta_2^2 \geq \|\mathbf{F}\|_F^2. \tag{12}$$

## Step 2: First bound on the loss

Denote $\mathcal{D} := \{\mathcal{D}(\mathbf{H}, \mathbf{H}_0)^{1/2} + \mathcal{D}(\mathbf{H}_0, \mathbf{H})^{1/2}\}$. By Assumption (**A2**) we have

$$\boldsymbol{z}_0 + \boldsymbol{U}B_{K-1}(\mu) \subseteq \text{conv}(\boldsymbol{X}_0) \subseteq \text{conv}(\boldsymbol{H}_0),$$

where $\boldsymbol{z}_0 + \boldsymbol{U}B_{K-1}(\mu)$ is a parametrization of the ball of center $\boldsymbol{z}_0$ and radius $\mu$ described in Assumption (**A2**) with $\boldsymbol{U}$ a matrix whose columns are $K-1$ orthonormal vectors. Using Lemma 25, we get that

$$\mu\sqrt{2} \leq \sigma_{\min}(\boldsymbol{H}_0) \leq \sigma_{\max}(\boldsymbol{H}_0),$$

where $\sigma_{\min}(\boldsymbol{H}_0), \sigma_{\max}(\boldsymbol{H}_0)$ denote its largest and smallest nonzero singular values. Then, since $\boldsymbol{z}_0 \in \text{conv}(\boldsymbol{H}_0)$ we have $\boldsymbol{z}_0 = \boldsymbol{H}_0^\top \alpha_0$ for some $\alpha_0$ s.t. $\alpha_0 \mathbf{1} = \mathbf{1}$. It holds,

$$\|\boldsymbol{z}_0\|_2 \leq \sigma_{\max}(\boldsymbol{H}_0)\|\alpha_0\|_2 \leq \sigma_{\max}(\boldsymbol{H}_0). \tag{13}$$

Note that

$$\sigma_{\max}(\widehat{\boldsymbol{H}} - \mathbf{1}\boldsymbol{z}_0^\top) \leq \sigma_{\max}(\widehat{\boldsymbol{H}}) + \sigma_{\max}(\mathbf{1}\boldsymbol{z}_0^\top) = \sigma_{\max}(\widehat{\boldsymbol{H}}) + \sqrt{K}\|\boldsymbol{z}_0\|_2. \tag{14}$$

Therefore, using Lemma 27 we have

$$\mathcal{D} \leq c\left[K^{3/2}\Delta_{1/2}\kappa(\boldsymbol{P}_0(\widehat{\boldsymbol{H}})) + \frac{\sigma_{\max}(\widehat{\boldsymbol{H}})\Delta_{1/2}K^{1/2}}{\mu} + \frac{K\Delta_{1/2}\|\boldsymbol{z}_0\|_2}{\mu}\right] + c\sqrt{K}\|\mathbf{F}\|_F, \tag{15}$$

where $\Delta_{1/2}$ equals $\Delta_1$ for problem (9) and $\Delta_2$ for problem (11), and $\kappa(\boldsymbol{A})$ stands for the conditioning number of matrix $\boldsymbol{A}$. In addition, Lemma 28 implies that

$$\mathcal{L}(\boldsymbol{H}_0, \widehat{\boldsymbol{H}})^{1/2} \leq \frac{1}{\alpha}\max\left\{(1+\sqrt{2})\sqrt{K}, \sqrt{2}\kappa(\boldsymbol{H}_0)\right\}\mathcal{D}. \tag{16}$$

## Step 3: Combining and final bound

By Lemma 29 it holds

$$\mathcal{D} \leq c\Big[\frac{K^{3/2}\mathcal{D}\Delta_{1/2}}{\alpha(\mu - 2\Delta_{1/2})\sqrt{2}} + \frac{K^2\sigma_{\max}(\boldsymbol{H}_0)\Delta_{1/2}}{(\mu - 2\Delta_{1/2})\sqrt{2}} + \frac{\mathcal{D}K^{1/2}\Delta_{1/2}}{\alpha\mu}$$
$$+ \frac{\sigma_{\max}(\boldsymbol{H}_0)\Delta_{1/2}K}{\mu} + \frac{K\Delta_{1/2}\|\boldsymbol{z}_0\|_2}{\mu}\Big] + c\sqrt{K}\|\mathbf{F}\|_F. \tag{17}$$

We understand that $\mathcal{D} = \mathcal{O}_{\Delta_{1/2} \to 0}(\Delta_{1/2})$ and for small enough $\Delta_{1/2}$ there exists a constant $c > 0$ such that

$$\mathcal{D} \le c\Delta_{1/2} + c\sqrt{K}\|\mathbf{F}\|_F$$

By (10) and (12), it yields that for small enough noise error $\|\mathbf{F}\|_F$ one has

$$\mathcal{D} \le c\|\mathbf{F}\|_F \,,$$

for some (other) constant $c > 0$. Plugging this result in (16) we prove the result.

## B.3  Proof of Corollary 18

Since the problem is convex in $\mathbf{W}$, the optimal solution $\hat{\mathbf{W}}$ is characterized by the first-order optimality condition. For any feasible $\mathbf{W}$ (satisfying $\mathbf{W} \ge 0, \mathbf{W}\mathbf{1} = 1$):

$$\langle \nabla_{\mathbf{W}}\mathcal{L}(\hat{\mathbf{W}}, \hat{\mathbf{H}}), \mathbf{W} - \hat{\mathbf{W}} \rangle \ge 0 \tag{18a}$$

where $\mathcal{L}(\mathbf{W}, \hat{\mathbf{H}}) = \frac{1}{2}\|\hat{\mathbf{N}} - \mathbf{W}\hat{\mathbf{H}}\|_F^2$ whose gradient is $\nabla_{\mathbf{W}}\mathcal{L} = (\mathbf{W}\hat{\mathbf{H}} - \hat{\mathbf{N}})\hat{\mathbf{H}}^\top$.

Substituting $\mathbf{W} = \mathbf{W}_0$ into the inequality for $\hat{\mathbf{W}}$:

$$\langle (\hat{\mathbf{W}}\hat{\mathbf{H}} - \hat{\mathbf{N}})\hat{\mathbf{H}}^\top, \mathbf{W}_0 - \hat{\mathbf{W}} \rangle \ge 0 \implies \langle (\hat{\mathbf{N}} - \hat{\mathbf{W}}\hat{\mathbf{H}})\hat{\mathbf{H}}^\top, \hat{\mathbf{W}} - \mathbf{W}_0 \rangle \ge 0 \tag{18b}$$

Furthermore, for any vectors $U, V$:

$$\langle (U\hat{\mathbf{H}} - V\hat{\mathbf{H}})\hat{\mathbf{H}}^\top, U - V \rangle = \|(U - V)\hat{\mathbf{H}}\|_F^2 \tag{18c}$$

We want to bound $\Delta\mathbf{W} = \hat{\mathbf{W}} - \mathbf{W}_0$. We can rewrite the observation as $\mathbf{X} = \mathbf{W}_0\mathbf{H}_0 + \mathbf{F}$. We consider the gradient at the true parameters $\mathbf{W}_0$ projected onto the difference $\Delta\mathbf{W} = \hat{\mathbf{W}} - \mathbf{W}_0$. We know that $\langle \nabla_{\mathbf{W}}\mathcal{L}(\hat{\mathbf{W}}), \hat{\mathbf{W}} - \mathbf{W}_0 \rangle \le 0$ and therefore by (18c):

$$\langle -\nabla_{\mathbf{W}}\mathcal{L}(\mathbf{W}_0), \hat{\mathbf{W}} - \mathbf{W}_0 \rangle \ge \langle \nabla_{\mathbf{W}}\mathcal{L}(\hat{\mathbf{W}}) - \nabla_{\mathbf{W}}\mathcal{L}(\mathbf{W}_0), \hat{\mathbf{W}} - \mathbf{W}_0 \rangle = \|(\hat{\mathbf{W}} - \mathbf{W}_0)\hat{\mathbf{H}}\|_F^2 \tag{18d}$$

Now we analyze the term $-\nabla_{\mathbf{W}}\mathcal{L}(\mathbf{W}_0)$. Using the definition of the gradient:

$$-\nabla_{\mathbf{W}}\mathcal{L}(\mathbf{W}_0) = -(\mathbf{W}_0\hat{\mathbf{H}} - \hat{\mathbf{N}})\hat{\mathbf{H}}^\top = (\hat{\mathbf{N}} - \mathbf{W}_0\hat{\mathbf{H}})\hat{\mathbf{H}}^\top$$

Crucially, $\hat{\mathbf{N}}$ is the completed matrix associated with the optimal solution $\hat{\mathbf{W}}$. By the definition, $\hat{\mathbf{N}}$ satisfies the constraints: $\mathcal{T}(\hat{\mathbf{N}}) = \mathbf{X}$ (consistency with observations); $\mathcal{T}^\perp(\hat{\mathbf{N}}) = \mathcal{T}^\perp(\hat{\mathbf{W}}\hat{\mathbf{H}})$ (optimality on missing entries minimizes the Frobenius norm). We decompose the residual $\hat{\mathbf{N}} - \mathbf{W}_0\hat{\mathbf{H}}$ using the mask projection $\mathcal{T}$:

$$\hat{\mathbf{N}} - \mathbf{W}_0\hat{\mathbf{H}} = \mathcal{T}(\hat{\mathbf{N}} - \mathbf{W}_0\hat{\mathbf{H}}) + \mathcal{T}^\perp(\hat{\mathbf{N}} - \mathbf{W}_0\hat{\mathbf{H}})$$
$$= \mathcal{T}(\mathbf{X} - \mathbf{W}_0\hat{\mathbf{H}}) + \mathcal{T}^\perp(\hat{\mathbf{W}}\hat{\mathbf{H}} - \mathbf{W}_0\hat{\mathbf{H}}) \,.$$

Substituting $\mathbf{X} = \mathbf{W}_0\mathbf{H}_0 + \mathbf{F}$:

$$\hat{\mathbf{N}} - \mathbf{W}_0\hat{\mathbf{H}} = \mathcal{T}(\mathbf{W}_0\mathbf{H}_0 + \mathbf{F} - \mathbf{W}_0\hat{\mathbf{H}}) + \mathcal{T}^\perp((\hat{\mathbf{W}} - \mathbf{W}_0)\hat{\mathbf{H}})$$
$$= \mathcal{T}(\mathbf{W}_0(\mathbf{H}_0 - \hat{\mathbf{H}}) + \mathbf{F}) + \mathcal{T}^\perp(\Delta\mathbf{W}\hat{\mathbf{H}}) \,.$$

We substitute this back into the inner product (18d) with $\Delta\mathbf{W}\hat{\mathbf{H}}$:

$$\langle (\hat{\mathbf{N}} - \mathbf{W}_0\hat{\mathbf{H}})\hat{\mathbf{H}}^\top, \Delta\mathbf{W} \rangle = \langle \hat{\mathbf{N}} - \mathbf{W}_0\hat{\mathbf{H}}, \Delta\mathbf{W}\hat{\mathbf{H}} \rangle$$
$$= \langle \mathcal{T}(\mathbf{W}_0(\mathbf{H}_0 - \hat{\mathbf{H}}) + \mathbf{F}) + \mathcal{T}^\perp(\Delta\mathbf{W}\hat{\mathbf{H}}), \mathcal{T}(\Delta\mathbf{W}\hat{\mathbf{H}}) + \mathcal{T}^\perp(\Delta\mathbf{W}\hat{\mathbf{H}}) \rangle$$

Since $\mathcal{T}$ and $\mathcal{T}^\perp$ are orthogonal projections ($\langle \mathcal{T}(A), \mathcal{T}^\perp(B) \rangle = 0$):

$$\langle -\nabla\mathcal{L}(\mathbf{W}_0), \Delta\mathbf{W} \rangle = \langle \mathcal{T}(\mathbf{W}_0(\mathbf{H}_0 - \hat{\mathbf{H}}) + \mathbf{F}), \mathcal{T}(\Delta\mathbf{W}\hat{\mathbf{H}}) \rangle + \|\mathcal{T}^\perp(\Delta\mathbf{W}\hat{\mathbf{H}})\|_F^2 \tag{18e}$$

From (18d) we deduce that

$$\|\Delta\mathbf{W}\hat{\mathbf{H}}\|_F^2 \leq \langle -\nabla\mathcal{L}(\mathbf{W}_0), \Delta\mathbf{W} \rangle$$
$$= \langle \mathcal{T}(\mathbf{W}_0(\mathbf{H}_0 - \hat{\mathbf{H}}) + \mathbf{F}), \mathcal{T}(\Delta\mathbf{W}\hat{\mathbf{H}}) \rangle + \|\mathcal{T}^{\perp}(\Delta\mathbf{W}\hat{\mathbf{H}})\|_F^2$$

Decomposing the LHS as $\|\Delta\mathbf{W}\hat{\mathbf{H}}\|_F^2 = \|\mathcal{T}(\Delta\mathbf{W}\hat{\mathbf{H}})\|_F^2 + \|\mathcal{T}^{\perp}(\Delta\mathbf{W}\hat{\mathbf{H}})\|_F^2$ and canceling $\|\mathcal{T}^{\perp}(\Delta\mathbf{W}\hat{\mathbf{H}})\|_F^2$ from both sides:

$$\|\mathcal{T}(\Delta\mathbf{W}\hat{\mathbf{H}})\|_F^2 \leq \langle \mathcal{T}(\mathbf{W}_0(\mathbf{H}_0 - \hat{\mathbf{H}}) + \mathbf{F}), \mathcal{T}(\Delta\mathbf{W}\hat{\mathbf{H}}) \rangle \tag{18f}$$

By Cauchy-Schwarz:

$$\|\mathcal{T}(\Delta\mathbf{W}\hat{\mathbf{H}})\|_F^2 \leq \|\mathcal{T}(\mathbf{W}_0(\mathbf{H}_0 - \hat{\mathbf{H}}) + \mathbf{F})\|_F \|\mathcal{T}(\Delta\mathbf{W}\hat{\mathbf{H}})\|_F \tag{18g}$$

Dividing by $\|\mathcal{T}(\Delta\mathbf{W}\hat{\mathbf{H}})\|_F$:

$$\|\mathcal{T}(\Delta\mathbf{W}\hat{\mathbf{H}})\|_F \leq \|\mathbf{W}_0(\mathbf{H}_0 - \hat{\mathbf{H}})\|_F + \|\mathbf{F}\|_F \leq \|\mathbf{W}_0\|_F \|\mathbf{H}_0 - \hat{\mathbf{H}}\|_F + \|\mathbf{F}\|_F \tag{18h}$$

Let $Z = \Delta\mathbf{W}\hat{\mathbf{H}}$. Since $\mathcal{T}$ acts as a block selector that preserves the training rows, we have:

$$\|\mathcal{T}(Z)\|_F \geq \|\mathcal{T}_T(Z)\|_F$$

From Lemma 25 and under the assumptions of Theorem 17, we know that $\sigma_{\min}(\hat{\mathbf{H}}_T) \geq c\mu$, hence

$$\|\mathcal{T}(\Delta\mathbf{W}\hat{\mathbf{H}})\|_F \geq \|\mathcal{T}_T(\Delta\mathbf{W}\hat{\mathbf{H}})\|_F \geq c\mu\|\Delta\mathbf{W}\|_F \,,$$

and we deduce the result by (18h).

## B.4 Proof of Theorem 20

We analyze the convergence using the Proximal Alternating Linearized Minimization (PALM) framework established in Bolte et al. (2014). We formulate the global objective function $\Psi(\mathbf{H}, \mathbf{W}, \mathbf{N})$ as the sum of a smooth coupling function and proper, lower semi-continuous, block-separable regularization terms:

$$\Psi(\mathbf{H}, \mathbf{W}, \mathbf{N}) := h(\mathbf{H}, \mathbf{W}, \mathbf{N}) + f(\mathbf{H}, \mathbf{N}) + g(\mathbf{W}), \tag{19}$$

where $h$ represents the smooth coupling term:

$$h(\mathbf{H}, \mathbf{W}, \mathbf{N}) := \frac{1}{2}\|\mathbf{N} - \mathbf{W}\mathbf{H}\|_F^2. \tag{20}$$

The regularization terms enforce the constraints as follows:

- $f(\mathbf{H}, \mathbf{N}) = \delta_{\geq 0}(\mathbf{H}) + p(\mathbf{N}) + \lambda\mathcal{D}(\mathbf{H}, \mathbf{N})$, where $\delta_{\geq 0}$ is the indicator function for non-negativity; and $p(\mathbf{N}) = \delta_{\mathcal{S}}(\mathbf{N})$, where $\mathcal{S} = \{\mathbf{Z} \in \mathbb{R}^{n \times p} \mid T(\mathbf{Z}) = \mathbf{X}\}$ is the affine set defined by the observation mask.

- $g(\mathbf{W}) = \delta_{\Delta}(\mathbf{W})$, the indicator function of the simplex (constraints $\mathbf{W} \geq 0, \mathbf{W}\mathbf{1} = \mathbf{1}$).

**Handling the Mask Constraint on $\mathbf{N}$.** The function $p(\mathbf{N})$ specifically addresses the block structure of $\mathbf{N}$. The set $\mathcal{S}$ constrains the observed blocks (where the mask is active) to equal the observation $\mathbf{X}$, while leaving the forecast blocks (where the mask is inactive) unconstrained. The proximal operator for $p(\mathbf{N})$ is the Euclidean projection onto the affine set $\mathcal{S}$, denoted as $\mathcal{P}_{\mathbf{X}}$:

$$\text{prox}_{\gamma, p}(\mathbf{U}) = \underset{\mathbf{N}}{\text{argmin}} \left( \frac{1}{2\gamma}\|\mathbf{N} - \mathbf{U}\|_F^2 + \delta_{\mathcal{S}}(\mathbf{N}) \right) = \mathcal{P}_{\mathbf{X}}(\mathbf{U}). \tag{21}$$

This operator fixes $\mathbf{N}_{jk} = \mathbf{X}_{jk}$ for observed entries and updates $\mathbf{N}_{jk} = \mathbf{U}_{jk}$ for missing/forecast entries.

**Algorithm Approximation and Analysis.** Strictly speaking, Algorithm 1 is an approximation of the exact PALM framework because the regularization term $f(\mathbf{H}, \mathbf{N})$ contains a coupling term $\lambda \mathcal{D}(\mathbf{H}, \mathbf{N})$ between blocks $\mathbf{H}$ and $\mathbf{N}$ that is not part of the smooth function $h$. Algorithm 1 addresses this by splitting the update of $\mathbf{H}$ into a gradient step on $h$ (Step 3) followed by a correction step using the archetypal projection (Step 5), and updating $\mathbf{N}$ via a projected gradient step on $h$ (Step 7). We analyze the convergence properties assuming this scheme approximates the minimization with respect $\mathbf{N}$ at each step. We could have made a loop over $\mathbf{N}$ to ensure convergence, at the price of computing time.

A key advantage of this formulation is that we can **get rid of N in the Lipschitz analysis** of the smooth term $h$. The partial gradients of $h$ are:

$$\nabla_{\mathbf{H}} h(\mathbf{H}, \mathbf{W}, \mathbf{N}) = \mathbf{W}^\top (\mathbf{W}\mathbf{H} - \mathbf{N}),$$
$$\nabla_{\mathbf{W}} h(\mathbf{H}, \mathbf{W}, \mathbf{N}) = (\mathbf{W}\mathbf{H} - \mathbf{N})\mathbf{H}^\top,$$
$$\nabla_{\mathbf{N}} h(\mathbf{H}, \mathbf{W}, \mathbf{N}) = \mathbf{N} - \mathbf{W}\mathbf{H}.$$

The Lipschitz constant for the partial gradient w.r.t $\mathbf{H}$, denoted $L_H(\mathbf{W})$, depends only on $\mathbf{W}$:

$$\|\nabla_{\mathbf{H}} h(\mathbf{H}_1, \mathbf{W}, \mathbf{N}) - \nabla_{\mathbf{H}} h(\mathbf{H}_2, \mathbf{W}, \mathbf{N})\|_F \leq \|\mathbf{W}^\top \mathbf{W}\|_F \|\mathbf{H}_1 - \mathbf{H}_2\|_F. \tag{22}$$

Similarly, $L_W(\mathbf{H})$ depends only on $\mathbf{H}$, and $L_N = 1$. Crucially, the variable $\mathbf{N}$ does not appear in the Lipschitz constants $L_H$ or $L_W$. This decouples the step-size requirements: the "free" blocks of $\mathbf{N}$ (forecasts) are updated via a projection $\mathcal{P}_{\mathbf{X}}$ that is contractive, and the smooth coupling $h$ ensures descent without requiring $\mathbf{N}$ to be bounded a priori in the gradient estimates.

Since $\Psi$ is semi-algebraic (composed of polynomial functions and indicator functions of semi-algebraic sets), it satisfies the Kurdyka-Łojasiewicz (KL) property. Following Theorem 1 in Bolte et al. (2014), and noting that the updates in Algorithm 1 ensure sufficient decrease of the objective $\Psi$, the sequence converges to a critical point.

# C   Propositions and Lemmas

• Results that we can use directly from Javadi & Montanari (2020b): Lemma B.1, Lemma B.2, Lemma B.3.

• Results of Javadi & Montanari (2020b) that has to be adapted: Lemme B.4 (done in Lemma 25), Lemma B.5 (done in Lemma 26), and Lemma B.6 (done in Lemma 27).

**Proposition 24** *For $\widehat{\boldsymbol{H}}$ solution to* (9) *(or* (11)*) one has* $\widetilde{\mathcal{D}}(\widehat{\boldsymbol{H}}, \mathbf{X}) \leq \widetilde{\mathcal{D}}(\mathbf{H}_0, \mathbf{X})$.

**Proof.** Observe that $\overline{\mathcal{D}}(\mathbf{X}, \mathbf{H}_0) = \|\mathbf{F}\|_F^2$. By (10) and (12), $\mathbf{H}_0$ is feasible for (9) (or (11)) then $\widetilde{\mathcal{D}}(\widehat{\boldsymbol{H}}, \mathbf{X}) \leq \widetilde{\mathcal{D}}(\mathbf{H}_0, \mathbf{X})$ ∎

**Lemma 25 (Adapted version of Lemma B.4 of Javadi & Montanari (2020b))** *If $\boldsymbol{H}$ is feasible for problem* (9) *(or* (11)*) and has linearly independent rows, then we have*

$$\min\{\sigma_{\min}(\boldsymbol{H}), \sigma_{\min}(\boldsymbol{H}_T)\} \geq \sqrt{2}(\mu - 2\Delta_{1/2}), \tag{23}$$

*where $\Delta_{1/2}$ equals $\Delta_1$ for problem* (9) *and $\Delta_2$ for problem* (11)*.*

**Proof.** Consider the notation and the outline of proof Lemma B.4 in Javadi & Montanari (2020b). The adaptation is simple using Assumption ($\mathbb{A}_2$). The trick is to only consider rows in the training set for $\sigma_{\min}(\boldsymbol{H})$, $\mathcal{T}_{\text{train}}(\mathbf{X}_0)$: the indice $i$ of proof of Lemma B.4 in Javadi & Montanari (2020b) correspond to the $n - N$ first rows in our case (the training set); and one should replace $\mathbf{X}_0$ by $\mathcal{T}_{\text{train}}(\mathbf{X}_0)$. This proof requires only feasibility of $\boldsymbol{H}$ and works no matter if a nonnegative constraint on $\boldsymbol{H}$ is active (as in Program (11)). As for $\sigma_{\min}(\boldsymbol{H}_T)$, we consider all the rows but only the $n - N$ first columns corresponding to the first and third blocks (selector operator $\mathcal{T}_T$). ∎

**Lemma 26 (Adapted version of Lemma B.5 of Javadi & Montanari (2020b))** *For $\widehat{\boldsymbol{H}}$ solution to* (9) *(or* (11)*), it holds*

$$\widetilde{\mathcal{D}}(\widehat{\boldsymbol{H}}, \mathbf{X}_0)^{1/2} \leq \widetilde{\mathcal{D}}(\mathbf{H}_0, \mathbf{X}_0)^{1/2} + c\sqrt{K}\|\mathbf{F}\|_F.$$

**Proof.** Consider the notation and the outline of proof Lemma B.5 in Javadi & Montanari (2020b). Note that Eq. (B.103) holds by Proposition 24. Form Eq. (B.104), the proof remains unchanged once one substitutes $\mathcal{D}$ by $\widetilde{\mathcal{D}}$. ∎

**Lemma 27 (Adapted version of Lemma B.6 of Javadi & Montanari (2020b))** *For $\widehat{\boldsymbol{H}}$ the optimal solution of problem* (9) *(or* (11)*), we have*

$$\alpha(\mathcal{D}(\widehat{\boldsymbol{H}}, \boldsymbol{H}_0)^{1/2} + \mathcal{D}(\boldsymbol{H}_0, \widehat{\boldsymbol{H}})^{1/2}) \leq c\left[K^{3/2}\Delta_{1/2}\kappa(\boldsymbol{P}_0(\widehat{\boldsymbol{H}})) + \frac{\Delta_{1/2}\sqrt{K}}{\mu}\sigma_{\max}(\widehat{\boldsymbol{H}} - \mathbf{1}\boldsymbol{z}_0^\top)\right] + c\sqrt{K}\|\mathbf{F}\|_F \tag{24}$$

*where $\boldsymbol{P}_0 : \mathbb{R}^d \to \mathbb{R}^d$ is the orthogonal projector onto $\mathrm{aff}(\boldsymbol{H}_0)$ (in particular, $\boldsymbol{P}_0$ is an affine map), and $\Delta_{1/2}$ equals $\Delta_1$ for problem* (9) *and $\Delta_2$ for problem* (11)*.*

**Proof.** Invoke the proof of Lemma B.6 in Javadi & Montanari (2020b) using the fact that $\widetilde{\mathcal{D}}(\mathbf{H}, \mathbf{X}) \leq \mathcal{D}(\mathbf{H}, \mathbf{X})$ and $\overline{\mathcal{D}}(\mathbf{X}, \mathbf{H}) \leq \mathcal{D}(\mathbf{X}, \mathbf{H})$. ∎

**Lemma 28** *Let $\boldsymbol{H}, \boldsymbol{H}_0$ be matrices with linearly independent rows. We have*

$$\mathcal{L}(\boldsymbol{H}_0, \boldsymbol{H})^{1/2} \leq \sqrt{2}\kappa(\boldsymbol{H}_0)\mathcal{D}(\boldsymbol{H}_0, \boldsymbol{H})^{1/2} + (1 + \sqrt{2})\sqrt{K}\mathcal{D}(\boldsymbol{H}, \boldsymbol{H}_0)^{1/2}, \tag{25}$$

*where $\kappa(\boldsymbol{A})$ stands for the conditioning number of matrix $\boldsymbol{A}$.*

**Proof.** See Lemma B.2 in Javadi & Montanari (2020b) ∎

**Lemma 29** *It holds*

$$\kappa(\boldsymbol{P}_0(\widehat{\boldsymbol{H}})) \leq \left[\frac{\mathcal{D}}{\alpha(\mu - 2\Delta_{1/2})\sqrt{2}} + \frac{K^{1/2}\sigma_{\max}(\boldsymbol{H}_0)}{(\mu - 2\Delta_{1/2})\sqrt{2}}\right].$$

**Proof.** The proof is given by Equations B.189-194 in Javadi & Montanari (2020b). ∎

# D   Algorithms for mNMF

In this section, we report the *Block Coordinate Descend* (BCD) Algorithm (see Algorithm 3) and the accelerated *Hierarchical Alternate Least Square* (HALS) for mNMF (see Algorithm 5), which is a generalization of Algorithm described in Gillis & Glineur (2012) to the matrix factorization with mask.

---

**Algorithm 3** BCD for mNMF

---

1: **Initialization**: choose $\mathbf{H}^0 \geq \mathbf{0}, \mathbf{W}^0 \geq \mathbf{0}$, and $\mathbf{N}^0 \geq \mathbf{0}$, set $i := 0$.
2: **while** stopping criterion is not satisfied **do**
3:      $\mathbf{H}^{i+1} := \text{update}(\mathbf{H}^i, \mathbf{W}^i, \mathbf{N}^i)$
4:      $\mathbf{W}^{i+1} := \text{update}(\mathbf{H}^{i+1}, \mathbf{W}^i, \mathbf{N}^i)$
5:      $\mathbf{N}^{i+1} := \text{update}(\mathbf{H}^{i+1}, \mathbf{W}^{i+1}, \mathbf{N}^i)$
6:      $i := i + 1$
7: **end while**

---

---

**Algorithm 4** ALS for mNMF

---

1: **Initialization**: choose $\mathbf{H}^0 \geq \mathbf{0}, \mathbf{W}^0 \geq \mathbf{0}$, set $\mathbf{N}^0 := \mathcal{P}_{\mathbf{X}}(\mathbf{H}^0 \mathbf{W}^0)$ and $i := 0$.
2: **while** stopping criterion is not satisfied **do**
3:      $\mathbf{H}^{i+1} := \min_{\mathbf{H} \geq 0} \|\mathbf{N}^i - \mathbf{W}^i \mathbf{H}\|_F^2$
4:      $\mathbf{W}^{i+1} := \min_{\mathbf{W} \geq 0, \mathbf{W}\mathbf{1}=\mathbf{1}} \|\mathbf{N}^i - \mathbf{W}\mathbf{H}^{i+1}\|_F^2$
5:      set $\mathbf{N}^{i+1} := \mathcal{P}_{\mathbf{X}}(\mathbf{W}^{i+1}\mathbf{H}^{i+1})$
6:      $i := i + 1$
7: **end while**

---

---

**Algorithm 5** accelerated HALS for mNMF

---

1: **Initialization**: choose $\mathbf{H}^0 \geq \mathbf{0}, \mathbf{W}^0 \geq \mathbf{0}$, *nonnegative rank* $K$, and $\alpha > 0$. Set $\mathbf{N}^0 = \mathcal{P}_{\mathbf{X}}(\mathbf{W}^0 \mathbf{H}^0)$, $\rho_{\mathbf{W}} := 1 + n(m+K)/(m(K+1))$, $\rho_{\mathbf{H}} := 1 + m(n+K)/(n(K+1))$, and $i := 0$.
2: **while** stopping criterion is not satisfied **do**
3:      $\mathbf{A} := \mathbf{N}\mathbf{H}^{i^\top}, \mathbf{B} := \mathbf{H}^i \mathbf{H}^{i^\top}$
4:      **for** $k \leq k_{\mathbf{W}} := \lfloor 1 + \alpha\rho_{\mathbf{W}} \rfloor$ **do**
5:        **for** $\ell \in [K]$ **do**
6:          $C_\ell := \sum_{j=1}^{\ell-1} W_j^{k+1} B_{j\ell} + \sum_{j=\ell+1}^{K} W_j^k B_{j\ell}$
7:          $W_\ell^k := \max(0, (A_\ell - C_\ell)/B_{\ell\ell})$
8:        **end for**
9:      $\mathbf{W}^{k_{\mathbf{W}}} := \mathcal{P}_\Delta(\mathbf{W}^{k_{\mathbf{W}}})$
10:      **end for**
11:      $\mathbf{N} := \mathcal{P}_{\mathbf{X}}(\mathbf{W}^{k_{\mathbf{W}}} \mathbf{H}^i)$
12:      $\mathbf{A} := \mathbf{W}^{k_{\mathbf{W}}} \mathbf{N}, \mathbf{B} := \mathbf{W}^{k_{\mathbf{W}}^\top} \mathbf{W}^{k_{\mathbf{W}}}$
13:      **for** $k \leq k_{\mathbf{H}} := \lfloor 1 + \alpha\rho_{\mathbf{H}} \rfloor$ **do**
14:        **for** $\ell \in [K]$ **do**
15:          $C_\ell := \sum_{j=1}^{\ell-1} H_j^{k+1} B_{j\ell} + \sum_{j=\ell+1}^{n} H_j^k B_{j\ell}$
16:          $H_\ell^k := \max(0, (A_\ell - C_\ell)/B_{\ell\ell})$
17:        **end for**
18:      **end for**
19:      $\mathbf{W}^{i+1} := \mathbf{W}^{k_{\mathbf{W}}}, \mathbf{H}^{i+1} := \mathbf{H}^{k_{\mathbf{H}}}$
20:      $\mathbf{N} := \mathcal{P}_{\mathbf{X}}(\mathbf{W}^{i+1}\mathbf{H}^{i+1})$
21:      $i := i + 1$
22: **end while**

---

# E    KKT conditions for mNMF

In this section we determine the KKT condition for mNMF problem, namely

$$\min_{\substack{\mathbf{W1}=\mathbf{1},\mathbf{W}\geq\mathbf{0}\\ \mathbf{H}\geq\mathbf{0}\\ \mathcal{T}(\mathbf{N})=\mathbf{X}}} \|\mathbf{N}-\mathbf{WH}\|_F^2 =: \mathcal{F}(\mathbf{N},\mathbf{W},\mathbf{H})\,. \tag{mNMF}$$

Let us introduce the dual variables $\mathbf{V}\geq\mathbf{0}$, $\mathbf{G}\geq\mathbf{0}$, $t\in\mathbb{R}^n$, and $\mathbf{Z}\in\mathrm{range}(\mathcal{T})$ so that $\mathcal{T}(\mathbf{Z})=\mathbf{Z}$. The Lagrangian of mNMF problem is

$$\mathcal{L}(\mathbf{N},\mathbf{W},\mathbf{H},\mathbf{V},\mathbf{G},t,\mathbf{Z}) = \mathcal{F}(\mathbf{N},\mathbf{W},\mathbf{H}) - \langle\mathbf{W},\mathbf{V}\rangle + \langle\mathbf{W1}_K - \mathbf{1}_N, t\rangle - \langle\mathbf{H},\mathbf{G}\rangle - \langle\mathbf{N}-\mathbf{X},\mathbf{Z}\rangle\,.$$

The KKT condition are the following:

$$\nabla_{\mathbf{N}}\mathcal{L} = \mathbf{N} - \mathbf{WH} - \mathbf{Z} = \mathbf{0} \qquad\qquad \Longleftrightarrow \mathcal{T}(\mathbf{N}-\mathbf{WH}) = \mathbf{Z} \tag{26}$$
$$\text{and } \mathcal{T}^{\perp}(\mathbf{N}-\mathbf{WH}) = \mathbf{0}$$

$$\nabla_{\mathbf{W}}\mathcal{L} = (\mathbf{WH}-\mathbf{N})\mathbf{H}^{\top} - \mathbf{V} - t\mathbf{1}_K^{\top} = \mathbf{0} \qquad \Longleftrightarrow \mathbf{V} = (\mathbf{WH}-\mathbf{N})\mathbf{H}^{\top} - t\mathbf{1}_K^{\top} \tag{27}$$

$$\nabla_{\mathbf{H}}\mathcal{L} = \mathbf{W}^{\top}(\mathbf{WH}-\mathbf{N}) - \mathbf{G} = \mathbf{0} \qquad\quad \Longleftrightarrow \mathbf{G} = \mathbf{W}^{\top}(\mathbf{WH}-\mathbf{N}) \tag{28}$$

$$\langle\mathbf{W},\mathbf{V}\rangle = \mathbf{0} \qquad\qquad\qquad\qquad \Longleftrightarrow \langle\mathbf{W},\nabla_{\mathbf{W}}\mathcal{F} - t\mathbf{1}_K^{\top}\rangle = \mathbf{0} \tag{29}$$

$$\langle\mathbf{H},\mathbf{G}\rangle = \mathbf{0} \qquad\qquad\qquad\qquad \Longleftrightarrow \langle\mathbf{H},\nabla_{\mathbf{H}}\mathcal{F}\rangle = \mathbf{0} \tag{30}$$

From the complementarity conditions (29), it follows:

$$W_{i,j} > 0 \Longrightarrow V_{i,j} = 0 \Longrightarrow t_i = -(\nabla_{\mathbf{W}}\mathcal{F})_{ij}\ \forall j$$

In order to compute $t_i$, we can select a row $W^{(i)}$, find any entry $W_{i,j} > 0$ and apply the previous formula. In the practical implementation phase, in order to make the estimation of $t_i$'s numerically more stable, we can adopt a slightly different strategy by averaging the values of $t_i$ computed per row entry $W_{i,j} > 0$.

