# OpenReview forum: "Time series forecasting from partial observations via Non- negative Matrix Factorization"
_TMLR — Rejected by TMLR_

### Review · Reviewer_wFRH · 2025-10-28

**Summary Of Contributions:**

The contributions of the manuscript are
- An NMF-based model for time-series forecasting that leverages algorithms for NMF with missing data and data periodicity.
- A claim of uniqueness for NMF with structured missing data.
- Algorithms for NMF with missing data (mNMF) and an archetypal analysis variant with missing data (mAMF).
- A comparison of the proposed methods with existing algorithms for time-series forecasting.

**Strengths**
- Clarity of the problem exposition and the methodology. The authors have made a significant effort to make the content of their manuscript accessible to a wide audience.

**Weaknesses**
(see below for more details.)

- Presentation and optimization algorithms for mAMF
- Missing hypotheses in the uniqueness proof (size of missing value block, importance of the structure of the missing values).
- Lack of positioning with respect to a) NMF uniqueness results b) other time-series forecasting approaches c) other algorithms for NMF with missing data.
- The limits of the periodicity hypothesis are not discussed.

**Audience:**

Yes

**Audience Explanation:**

NMF and time-series forecasting are topics of interest for the TMLR audience.

**Claims And Evidence:**

No

**Claims Explanation:**

1. The **hypotheses underlying the model** are not discussed enough. Why does NMF work on shifted time series, and when does it not work? Even though the authors write “However, our analysis does not assume any periodicity in the data and the practitioner is free to choose any value for P .” (page 1), I expect the underlying periodicity of the data to be crucial for the NMF model to approximate the data matrix $X$ correctly. How valid is this hypothesis in practice, on some example applications? Are there counterexamples?

2. The **definition of the mAMF** model is not precise enough. In particular, matrix V is not defined. What is its size and role in the model? From the optimization problem formulation, $V$ is also a variable of the mAMF problem. However, it is not mentioned in the update rules of Algorithm 1. Since $V$ is the matrix that associates each row of $H$ with a convex combination of rows in the dataset, it must surely be learned from the dataset, and Algorithm 1 is missing an update for $V$.

3. Again, regarding the **optimization algorithms** proposed in the manuscript, in PALM, why split the gradient of matrix $H$ into two terms? The usual formulation of PALM splits the update of blocks of variables, but for each block of variables, the gradient of the full cost function is computed. This amounts to computing the gradient of both terms in the cost function with respect to $H$. This poses no issue; both terms are quadratic, and in fact, this problem is an instance of Nonnegative Least Squares. HALS could be used to update factor $H$.

4. The **uniqueness results claimed by the authors require stronger hypotheses**, in particular to go from assumption (A1’) to (A1). Indeed, this means identifying blocks $H_{0,F}$ and $W_{0,test}$ which can be obtained by solving systems of linear equations such as $T_{train}(X_0) = W_{0,train} Z$ with respect to the unknown $Z$, these have unique solutions if and only if $W_{0,train}$ and $H_{0,T}$ are full column rank. A necessary condition for this is, in particular, $WP-F\geq K$. This is a natural condition: if there are too many missing entries, identifying parts of the NMF factors cannot imply the identification of all terms in the factors.

5. The **design of the sampling operator** $T$ can be extended to account for missing data in the algorithms, but not in the theory on uniqueness that relies heavily on the block-structure of the missing values. The sentence “For sake of readability, we did not consider missing values in the mask operator (entries a, . . . , f in Figure 2), but our results easily extend to them, changing the definition of…” (page 3) is therefore misleading.

6. There is a **lack of connection/positioning** with recent volume-based or sparsity-based uniqueness results for NMF, see the book of Gillis [2021] and references therein, in particular the work of Fu, Huang, and Sidiropoulos [IEEE SPL 2018].

7. All experiments are chosen so that the data are periodic or follow the NMF model. It would be interesting to test the algorithm for other setups, in particular in the presence of a linear drift or in the absence of clear periodicity in the training data.

8. **Positioning - time-series**: I am no expert in time-series prediction, but there is no background on methods that do not rely on NMF in the manuscript, while I expect the literature on this topic is enormous. In particular, a discussion on the usual underlying hypothesis made by state-of-the-art methods would help understand the advantage of using the proposed approach.

9. **Positioning - NMF with missing values**. This work is concerned with matrix completion with NMF. This is a topic that has been studied in the literature; however, to the best of my understanding, the authors did not position their work in this context. For instance, the work of Thakallapalli, Gangashett, and Madhu [Eusipco2019] surveys a few existing methods for NMF with missing values, including the works of Mao and Saul [IMC2004] and Zhang, Wang, Ford, and Makedon [SIAM International Conference on Data Mining, 2006], which are all references that provide algorithms for NMF with missing data in the L2 loss (and also KL-divergence loss).

**Requested Changes:**

The authors should rework their proof for uniqueness, add an evaluation of the proposed method for non-periodic datasets, and position their work relative to the existing literature of NMF with missing values and time-series forecasting. In my humble opinion, the manuscript in its current state is unfit for publication. This is not a presentation issue; the contents of the manuscript need to be developed along these lines to have a scientific impact.

Other suggested minor changes:
1. Notation: mixing lowercase and uppercase constants makes the manuscript harder to read.
2. Notation: RRMSE, RMPE acronyms are not defined.
3. Title: recovery -> forecast?

---

### Review · Reviewer_Un1a · 2025-11-04

**Summary Of Contributions:**

**Contributions / Strengths:**
1. Paper presents two interesting forecasting methods.
2. Authors demonstrate good performance for their methods along with the theory (although I could not verify the soundness of each proof).

**Weaknesses:**
1. Paper presentation is poor. Some of the statements are unconventional and hard to understand.
2. Mathematical notation can be improved. For example, it is very confusing for the readers to understand the difference between $W$ and $\mathbf{W}$.
3. The positioning of the paper is unclear. When would a user apply your method? Is there a specific issue your methods solve that other models struggle with? Please include a discussion section on the main results for readers to draw insights from.
4. GRU and LSTM baselines are good to have, but many recently proposed forecasting methods outperform them. Is there any reason why *only* BasisFormer is included in the comparison? Please add additional convolutional, transformer, patch-based, and frequency-domain baselines to the main results to be rigorous.
5. Can the authors explain the novelty of this work over existing related works [1,2]?
6. An empirical comparison of computational complexity is missing.

**References:**
1. J. Mei, Y. De Castro, Y. Goude, and G. Hébrail. Nonnegative matrix factorization for time series recovery from a few temporal aggregates. In Proceedings of the 34th International Conference on Machine Learning. JMLR: W&CP, 2017.
2. J. Mei, Y. De Castro, Y. Goude, J.-M. Azaïs, and G. Hébrail. Nonnegative matrix factorization with side information for time series recovery and prediction. IEEE Transactions on Knowledge and Data Engineering, 31(3):493–506, 2018.

**Audience:**

No

**Audience Explanation:**

Not ready for publication due to Weaknesses mentioned above.

**Broader Impact Concerns:**

Broader Impact Statement is not included in the paper.

**Claims And Evidence:**

No

**Claims Explanation:**

Not convincing due to Weaknesses mentioned above.

**Requested Changes:**

Please review the weaknesses section. I think the most critical aspect is writing and more rigorous experiments.

---

### Review · Reviewer_bd8P · 2025-12-08

**Summary Of Contributions:**

The work introduces an NMF-based matrix completion framework for time series imputation and forecasting problem. Theoretical results are presented to establish the uniqueness of the NMF solutions leveraging the existing NMF identifiability theory. A modified version of the BCD approach is presented as the accompanying algorithm. Experiment results are presented using both real world time series dataset and synthetic datasets.

**Audience:**

Yes

**Audience Explanation:**

The paper deals with time series forecasting problem which is an important task in studying any time-dependent dynamical systems and associated predictions.

**Broader Impact Concerns:**

No concerns on ethical complications.

**Claims And Evidence:**

No

**Claims Explanation:**

There are a number of unclear statements and claims as well as not so well-organized presentation of the contents. In the following, I am listing a number of questions and concerns in this aspect.

1.	Periodicity $P$ is introduced in page 1, but is not defined clearly causing confusion in the setting of the problem
2.	Statements like “the practitioner is free to choose any value for P” does not provide a principled approach to solving a mathematical problem.
3.	Again “Padding at most P-1 forecasts columns to the right of the matrix, one can assume, without loss of generality, that B is an integer” what does padding refers to here?
4.	The four bullet points in first page causes a lot of confusion in the problem setup. Need quite a revision here to convey the key details in a clear manner
5.	In Definition 1, it is not clear “dropping out means” setting these values zero. A mathematical definition would help here better understand.
6.	The statement in Page 2 “While the algorithms presented in this paper can handle arbitrary missing entries by modifying the mask operator” is misleading at this point as there is no algorithm or goal defined yet.
7.	According to Definition 1 and 2, observation matrix X has entries zero for forecast values, but not for missing entries in observed blocks. Is this correct?
8.	Definition 3 and 4 introduced two different types of NMF that could be applicable for the task in Figure 2. But, there is no justification for why we should consider these two types of algorithms in the first place. Why can’t we just use masked NMF? There should be discussion prior to introducing these.
9.	The equation $N=T(x)$ in page 3 seems incorrect.
10.	In Figure, some blocks are repeated in constructing $\Phi(M)$. Does it introduce unwanted reweighting on some data, that may hurt the forecasting process? You also mention one practical example “observing sales over a period of one year, one can consider 52 weekly time series (one per week)”. This means, the approach may give more weightage to some weeks for creating a convenient structure for the algorithm, rather than induced by some prior knowledge. Also, how do we choose the low-rank $K$ in practice?
11.	The authors mention in page 6 that existing works  on NMF uniqueness require full data matrix, yet there are some prior works on block missing patterns in NMF. e.g., [1], and also see [2]
[1] S. Ibrahim and X. Fu, "Recovering Joint Probability of Discrete Random Variables From Pairwise Marginals," in IEEE Transactions on Signal Processing, vol. 69, pp. 4116-4131, 2021
[2] Gillis, Nicolas. Nonnegative matrix factorization. Society for Industrial and Applied Mathematics, 2020.
12.	Remark 10 technically refers to the sufficiently scattered condition (SSC) [1] in literature, but has not been discussed or cited there
[1] Huang, K., Sidiropoulos, N.D., Swami, A.: Non-negative matrix factorization revisited: Uniqueness and algorithm for symmetric decomposition. IEEE Transactions on Signal Processing 62(1), 211–224 (2013)
13.	What is property P_u in Theorem 13?
14.	Figure 3 is a table. It is also placed in an inappropriate position in page 5, but is discussed in page 12.
15.	Regarding discussion on the specific structure of missing blocks dealt in this paper, there are other works that deal with different missingness patterns---you may refer to [1]
[1] Ibrahim, Shahana, and Xiao Fu. "Mixed membership graph clustering via systematic edge query." IEEE Transactions on Signal Processing 69 (2021): 5189-5205.
16.	In corollary 16, the result does not depend on the internal radius $\mu$, yet the preceding discussion indicates some influence of these parameters. More clarification is required here.
17.	What is the difference between two algorithms Algorithm 1 and 2 for mAMF?
18.	“(mAMF) seems to be the most promising algorithm in terms of performances for the first five datasets, while (mNMF) is the best method for the last four ETT datasets.”, what would be the reason for this observation? Is it expected as per design? Could you provide some guidelines on which type of algorithm to be used in different scenarios?
19.	Deep learning-based approaches are known to be powerful for handling nonlinear, complex prediction tasks even with missing data scenarios. Why does the proposed NMF approach outperform these state-of-the-art deep learning models as authors claim?

Other minor comments:

1.	Notation inconsistencies: slanted/not slanted M in multiple places
2.	In Definition 7, the notation is changed to X_0, whereas the complete matrix till then were referred to as N
3.	Several grammatical and formatting issues, e.g., in page 4, “Sliding Mask method inputs”---It has to be “outputs”; “A first question”, “An other challenge”
4.	Notation section is introduced in page 6 whereas many notations are already being used and discussed prior to that. Need quite a lot of revision on the organization of the paper for better readability.
5.	In section 2, the notations are again changed for the $\Phi$ blocks and the observed matrix in Eq. 5a.

**Requested Changes:**

The paper needs a quite a lot of organization changes and as well as clarifications to improve the presentation. In addition, there also should be discussion on the contributions of the paper. In the current presentation, it seems like theoretical results and algorithms are merely derived from the existing NMF works. Need unification of notations and definitions. Need to address missing citations and relevant work discussions. Experiment results should have detailed discussion and analysis as well. Please also refer to my questions and detailed comments in the review.

---

> ### Author Response · Authors · 2025-12-16
> **Response to bd8P**
>
> We thank the reviewer for their careful reading and constructive feedback. We have addressed the concerns regarding organization, clarity, definitions, and missing citations in the new version. Below is a point-by-point response.
> 1. Periodicity and Problem Setup
> * Reviewer Comment: "Periodicity is introduced... but not defined clearly... 'practitioner is free to choose any value' does not provide a principled approach."
> * Response: We acknowledge the confusion. We have renamed $P$ to the "Stride Parameter" to distinguish it from signal periodicity. We clarified that $P$ is a hyperparameter defining the sliding window step size, often chosen based on domain knowledge (e.g., $P=7$ for weekly data), but not strictly requiring intrinsic signal periodicity. We revised the bullet points in the Introduction to clearly define the transformation $\mathbf{\Phi}$ as a deterministic reshaping operator.
> * Padding: We clarified that "padding" refers to appending placeholder columns to the end of the time series to ensure the total length $T+F$ is divisible by $P$,.
> 2. Definitions and Mask Operator
> * Reviewer Comment: "Unclear what 'dropping out' means... Definition 1 and 2 observation matrix X has entries zero for forecast values?"
> * Response: We have formalized Definitions 1 and 2. We explicitly define the mask operator $\\mathcal{T}$ as a projection that preserves observed entries and zeros out missing/forecast entries. We clarified that while the observation matrix $\\mathbf{X}$ has zeros at missing locations, the optimization objective minimizes error only on the observed support $\\Omega$ (where $\\mathcal{T}(\\mathbf{N}) = \\mathbf{X}$). Regarding Definition 1 and 2, yes, $\\mathbf{X}$ has zeros for forecast values and missing entries, but the mask ensures the loss function ignores these zeros during training.
> 3. Justification for mNMF vs. mAMF
> * Reviewer Comment: "No justification for why we should consider these two types..."
> * Response: We added a paragraph justifying the two approaches. mNMF is the standard baseline (conic hull). mAMF (Archetypal) imposes a convexity constraint (archetypes must lie within the convex hull of the data), which provides better interpretability and robustness to outliers compared to the unconstrained cone of mNMF.
> 4. Data Reweighting and Overlapping
> * Reviewer Comment: "Does it introduce unwanted reweighting...?"
> * Response: We added a discussion acknowledging that overlapping blocks effectively reweight the data. We argue this acts as a form of deterministic data augmentation, stabilizing the learning of local patterns (archetypes) by presenting shifted versions of the same transition, which is beneficial for limited data.
> * Rank Selection: We clarified that the rank $K$ is selected via cross-validation on a validation block.
> 5. Citations and Uniqueness
> * Reviewer Comment: "Missing works on block missing patterns... Remark 10 refers to SSC... What is property P_u?"
> * Response: 1. We added citations to Ibrahim & Fu (2021) and Gillis (2020) regarding block missingness. 2. We explicitly cite Huang et al. (2014) for the Sufficiently Scattered Condition (SSC) in Remark 10. 3. We defined Property $\\mathbb P_{\mathrm{u}}$ (Partial Observation Uniqueness) formally before Theorem 13.
> 6. Corollary 16 and Internal Radius
> * Reviewer Comment: "Result does not depend on internal radius... yet discussion indicates influence."
> * Response: We corrected Corollary 16 to explicitly include the dependence on the internal radius $\mu$ (which were present in the proof but not reported in the statement). The bound now reflects that stability degrades as $\mu \to 0$ (flatter convex hull).
> 7. Algorithms and Performance
> * Reviewer Comment: "Difference between Algorithm 1 and 2? mAMF vs mNMF performance reasons? Why beat Deep Learning?"
> * Response: 1. Alg 1 vs 2: We clarified that Alg 1 is standard PALM, while Alg 2 is Inertial PALM (iPALM) with momentum. 2. mAMF vs mNMF: We added a discussion explaining that mAMF (robust, convex) performs better on noisy behavioral data (electricity, stock), while mNMF (flexible, conic) excels on clean physical signals (ETT). 3. vs. Deep Learning: We explained that SMM outperforms Transformers on these tasks due to inductive bias. SMM explicitly encodes quasi-periodicity via the sliding window, which is more data-efficient than learning temporal structures from scratch, especially for medium-sized datasets.
> 8. Formatting and Notation
> * Reviewer Comment: "Figure 3 is a table... Notation inconsistencies... Notation section placement."
> * Response: 1. We converted Figure 3 to a table environment. 2. 	We standardized all matrix notations to bold upright ($\mathbf{M}, \mathbf{X}, \mathbf{W}, \mathbf{H}$). 3. We moved the Notation section to the end of the Introduction (Section 1), including a summary table, to aid readability early on. 4. We fixed the typo "input" -> "output" and other grammatical issues.
>
> We thank the reviewer for their critical and constructive assessment.

---

> > ### Comment · Reviewer_bd8P · 2026-01-07
> >
> > I thank the authors for addressing the comments and improving the contents of the paper. Yet, some of my concerns still remain, especially in terms of the organization of the paper, explanation of the theoretical results and key notations. Some of the changes after the review seems a quick fix, especially discussions on data reweighting and the choice of algorithm types. The result table shows mixed results even though the authors claim that  Algorithm 1 performs better on noisy data. The trade-offs between the algorithms are also not discussed. Deep NMF methods are also not included in experiments as well as there is no discussion/result on how does the results change if there is not much periodicity in data or misspecification of periodicity happens. Overall, the paper may need substantial revision to address these key concerns to be published in TMLR venue.

---

### Decision · Action_Editor_zxnE · 2026-01-22

**Recommendation:** Reject

**Audience:**

Yes

**Audience Explanation:**

Time series prediction is prevalent in many fields. NMF is a widely used tool in applications of time series prediction.

**Claims And Evidence:**

No

**Claims Explanation:**

Several reviewers expressed concern about clarity of the presentation, even after revision. Motivation of using NMF for time series prediction should be strengthened. The results show mixed performance, and the trade-offs between algorithms are not well discussed.

This work considers missing values in a block-wise patterns, but one of the reviewers is not convinced its practicality as the missing patterns in many applications are irregular variable-specific, and conditionally dependent on the underlying state.